# A Simple Unified Information Regularization Framework for Multi-Source Domain Adaptation

## Abstract

Adversarial learning strategy has demonstrated remarkable performance in dealing with single-source unsupervised Domain Adaptation (DA) problems, and it has recently been applied to multi-source DA problems. Although most existing DA methods use multiple domain discriminators, the effect of using multiple discriminators on the quality of latent space representations has been poorly understood. Here we provide theoretical insights into potential pitfalls of using multiple domain discriminators: First, domain-discriminative information is inevitably distributed across multiple discriminators. Second, it is not scalable in terms of computational resources. Third, the variance of stochastic gradients from multiple discriminators may increase, which significantly undermines training stability. To fully address these issues, we situate adversarial DA in the context of information regularization. First, we present a unified information regularization framework for multi-source DA. It provides a theoretical justification for using a single and unified domain discriminator to encourage the synergistic integration of the information gleaned from each domain. Second, this motivates us to implement a novel neural architecture called a Multi-source Information-regularized Adaptation Networks (MIAN). The proposed model significantly reduces the variance of stochastic gradients and increases computational-efficiency. Large-scale simulations on various multi-source DA scenarios demonstrate that MIAN, despite its structural simplicity, reliably outperforms other state-of-the-art methods by a large margin especially for difficult target domains.

## 1 Introduction

Although a large number of studies have demonstrated the ability of deep neural networks to solve challenging tasks, the tasks solved by networks are mostly confined to a similar type or a single domain. One remaining challenge is the problem known as domain shift (Gretton et al. (2009)), where a direct transfer of information gleaned from a single source domain to unseen target domains may lead to significant performance impairment. Domain adaptation (DA) approaches aim to mitigate this problem by learning to map data of both domains onto a common feature space. Whereas several theoretical results (Ben-David et al. (2007); Blitzer et al. (2008); Zhao et al. (2019a)) and algorithms for DA (Long et al. (2015; 2017); Ganin et al. (2016)) have focused on the case in which only a single-source domain dataset is given, we consider a more challenging and generalized problem of knowledge transfer, referred to as Multi-source unsupervised DA (MDA). Following a seminal theoretical result on MDA (Blitzer et al. (2008); Ben-David et al. (2010)), technical advances have been made, mainly on the adversarial methods. (Xu et al. (2018); Zhao et al. (2019c)).

While most of adversarial MDA methods use multiple independent domain discriminators (Xu et al. (2018); Zhao et al. (2018); Li et al. (2018); Zhao et al. (2019c;b)), the potential pitfalls of this setting have not been fully explored. The existing works do not provide a theoretical guarantee that the unnecessary domain-specific information is fully filtered out, because the domain-discriminative information is inevitably distributed across multiple discriminators. For example, the multiple domain discriminators focus only on estimating the domain shift between source domains and the target, while the discrepancies between the source domains are neglected, making it hard to align all the given domains. This necessitates garnering the domain-discriminative information with a

unified discriminator. Moreover, the multiple domain discriminator setting is not scalable in terms of computational resources especially when large number of source domains are given, e.g., medical reports from multiple patients. Finally, it may undermine the stability of training, as earlier works solve multiple independent adversarial minimax problems.

To overcome such limitations, we propose a novel MDA method, called Multi-source Information-regularized Adaptation Networks (**MIAN**), that constrains the mutual information between latent representations and domain labels. First, we show that such mutual information regularization is closely related to the explicit optimization of the $\mathcal{H}$-divergence between the source and target domains. This affords the theoretical insight that the conventional adversarial DA can be translated into an information-theoretic-regularization problem. Second, based on our findings, we propose a new optimization problem for MDA: minimizing adversarial loss over multiple domains with a single domain discriminator. We show that the domain shift between each source domain can be indirectly penalized, which is known to be beneficial in MDA (Li et al. (2018); Peng et al. (2019)), with a single domain discriminator. Moreover, by analyzing existing studies in terms of information regularization, we found that the variance of the stochastic gradients increases when using multiple discriminators.

Despite its structural simplicity, we found that **MIAN** works efficiently across a wide variety of MDA scenarios, including the DIGITS-Five (Peng et al. (2019)), Office-31 (Saenko et al. (2010)), and Office-Home datasets (Venkateswara et al. (2017)). Intriguingly, **MIAN** reliably and significantly outperformed several state-of-the-art methods that either employ a domain discriminator separately for each source domain (Xu et al. (2018)) or align the moments of deep feature distribution for every pairwise domain (Peng et al. (2019)).

## 2 RELATED WORKS

Several DA methods have been used in attempt to learn domain-invariant representations. Along with the increasing use of deep neural networks, contemporary work focuses on matching deep latent representations from the source domain with those from the target domain. Several measures have been introduced to handle domain shift, such as maximum mean discrepancy (MMD) (Long et al. (2014; 2015)), correlation distance (Sun et al. (2016); Sun & Saenko (2016)), and Wasserstein distance (Courty et al. (2017)). Recently, adversarial DA methods (Ganin et al. (2016); Tzeng et al. (2017); Hoffman et al. (2017); Saito et al. (2018; 2017)) have become mainstream approaches owing to the development of generative adversarial networks (Goodfellow et al. (2014)). However, the abovementioned single-source DA approaches inevitably sacrifice performance for the sake of multi-source DA.

Some MDA studies (Blitzer et al. (2008); Ben-David et al. (2010); Mansour et al. (2009); Hoffman et al. (2018)) have provided the theoretical background for algorithm-level solutions. (Blitzer et al. (2008); Ben-David et al. (2010)) explore the extended upper bound of true risk on unlabeled samples from the target domain with respect to a weighted combination of multiple source domains. Following these theoretical studies, MDA studies with shallow models (Duan et al. (2012b;a); Chattopadhyay et al. (2012)) as well as with deep neural networks (Mancini et al. (2018); Peng et al. (2019); Li et al. (2018)) have been proposed. Recently, some adversarial MDA methods have also been proposed. Xu et al. (2018) implemented a k-way domain discriminator and classifier to battle both domain and category shifts. Zhao et al. (2018) also used multiple discriminators to optimize the average case generalization bounds. Zhao et al. (2019c) chose relevant source training samples for the DA by minimizing the empirical Wasserstein distance between the source and target domains. Instead of using separate encoders, domain discriminators or classifiers for each source domain as in earlier works, our approach uses unified networks, thereby improving resource-efficiency and scalability.

Several existing MDA works have proposed methods to estimate the source domain weights following (Blitzer et al. (2008); Ben-David et al. (2010)). Mansour et al. (2009) assumed that the target hypothesis can be approximated by a convex combination of the source hypotheses. (Peng et al. (2019); Zhao et al. (2018)) suggested ad-hoc schemes for domain weights based on the empirical risk of each source domain. Li et al. (2018) computed a softmax-transformed weight vector using the empirical Wasserstein-like measure instead of the empirical risks. Compared to the proposed methods without robust theoretical justifications, our analysis does not require any assumption or estimation for the domain coefficients. In our framework, the representations are distilled to be independent of the domain, thereby rendering the performance relatively insensitive to explicit weighting strategies.

# 3 THEORETICAL INSIGHTS

We first introduce the notations for the MDA problem in classification. A set of source domains and the target domain are denoted by $\{D_{S_i}\}_{i=1}^N$ and $D_T$, respectively. Let $X_{S_i} = \left\{\mathbf{x}_{S_i}^j\right\}_{j=1}^m$ and $Y_{S_i} = \left\{\mathbf{y}_{S_i}^j\right\}_{j=1}^m$ be a set of $m$ i.i.d. samples from $D_{S_i}$. Let $X_T = \left\{\mathbf{x_T}^j\right\}_{j=1}^m \sim (D_T^X)^m$ be the set of $m$ i.i.d. samples generated from the marginal distribution $D_T^X$. The domain label and its probability distribution are denoted by $V$ and $P_V(\mathbf{v})$, where $\mathbf{v} \in \mathcal{V}$ and $\mathcal{V}$ is the set of domain labels. In line with prior works (Hoffman et al. (2012); Gong et al. (2013); Mancini et al. (2018); Gong et al. (2019)), domain label can be generally treated as a stochastic latent random variable in our framework. However, for simplicity, we take the empirical version of the true distributions with given samples assuming that the domain labels for all samples are known. The latent representation of the sample is given by $Z$, and the encoder is defined as $F : \mathcal{X} \to \mathcal{Z}$, with $\mathcal{X}$ and $\mathcal{Z}$ representing data space and latent space, respectively. Accordingly, $Z_{S_i}$ and $Z_T$ refer to the outputs of the encoder $F(X_{S_i})$ and $F(X_T)$, respectively. For notational simplicity, we will omit the index $i$ from $D_{S_i}$, $X_{S_i}$ and $Z_{S_i}$ when $N = 1$. A classifier is defined as $C : \mathcal{Z} \to \mathcal{Y}$ where $\mathcal{Y}$ is the class label space.

## 3.1 PROBLEM FORMULATION

For comparison with our formulation, we recast single-source DA as a constrained optimization problem. The true risk $\epsilon_T(h)$ on unlabeled samples from the target domain is bounded above the sum of three terms (Ben-David et al. (2010)): (1) true risk $\epsilon_S(h)$ of hypothesis $h$ on the source domain; (2) $\mathcal{H}$-divergence $d_{\mathcal{H}}(D_S, D_T)$ between a source and a target domain distribution; and (3) the optimal joint risk $\lambda^*$.

**Theorem 1** (Ben-David et al. (2010)). *Let hypothesis class $\mathcal{H}$ be a set of binary classifiers $h : \mathcal{X} \to \{0, 1\}$. Then for the given domain distributions $D_S$ and $D_T$,*

$$\forall h \in \mathcal{H}, \epsilon_T(h) \leq \epsilon_S(h) + d_{\mathcal{H}}(D_S, D_T) + \lambda^*, \tag{1}$$

*where $d_{\mathcal{H}}(D_S, D_T) = 2 \sup_{h \in \mathcal{H}} \left| \mathbb{E}_{\mathbf{x} \sim D_S^X} \left[ \mathbb{I}(h(\mathbf{x}) = 1) \right] - \mathbb{E}_{\mathbf{x} \sim D_T^X} \left[ \mathbb{I}(h(\mathbf{x}) = 1) \right] \right|$ and $\mathbb{I}(a)$ is an indicator function whose value is $1$ if $a$ is true, and $0$ otherwise.*

The empirical $\mathcal{H}$-divergence $\hat{d}_{\mathcal{H}}(X_S, X_T)$ can be computed as follows (Ben-David et al. (2010)):

**Lemma 1.**

$$\hat{d}_{\mathcal{H}}(X_S, X_T) = 2 \left( 1 - \min_{h \in \mathcal{H}} \left[ \frac{1}{m} \sum_{\mathbf{x} \in X_S} \mathbb{I}[h(\mathbf{x}) = 1] + \frac{1}{m} \sum_{\mathbf{x} \in X_T} \mathbb{I}[h(\mathbf{x}) = 0] \right] \right) \tag{2}$$

Following Lemma 1, a domain classifier $h : \mathcal{Z} \to \mathcal{V}$ can be used to compute the empirical $\mathcal{H}$-divergence. Suppose the optimal joint risk $\lambda^*$ is sufficiently small as assumed in most adversarial DA studies (Saito et al. (2017); Chen et al. (2019)). Thus, one can obtain the ideal encoder and classifier minimizing the upper bound of $\epsilon_T(h)$ by solving the following min-max problem:

$$
\begin{aligned}
F^*, C^* &= \arg\min_{F,C} \quad L(F, C) + \beta \hat{d}_{\mathcal{H}}(Z_S, Z_T) \\
&= \arg\min_{F,C} \max_{h \in \mathcal{H}} \quad L(F, C) + \beta \frac{1}{m} \Big( \sum_{i:\mathbf{z}_i \in Z_S} \mathbb{I}[h(\mathbf{z}_i) = 1] + \sum_{j:\mathbf{z}_j \in Z_T} \mathbb{I}[h(\mathbf{z}_j) = 0] \Big),
\end{aligned}
\tag{3}
$$

where $L(F, C)$ is the loss function on samples from the source domain, $\beta$ is a Lagrangian multiplier, $\mathcal{V} = \{0, 1\}$ such that each source instance and target instance are labeled as 1 and 0, respectively, and $h$ is the binary domain classifier. Note that the latter min–max problem is obtained by converting $-min$ into $max$ and removing the constant term from Lemma 1.

## 3.2 INFORMATION-REGULARIZED MIN–MAX PROBLEM FOR MDA

Intuitively, it is not highly desirable to adapt the learned representation in the given domain to the other domains, particularly when the representation itself is not sufficiently domain-independent.

This motivates us to explore ways to learn representations independent of domains. Inspired by a contemporary fair model training study (Roh et al. (2020)), the mutual information between the latent representation and the domain label $I(Z; V)$ can be expressed as follows:

**Theorem 2.** *Let $P_Z(\mathbf{z})$ be the distribution of $Z$ where $\mathbf{z} \in \mathcal{Z}$. Let $h$ be a domain classifier $h : \mathcal{Z} \to \mathcal{V}$, where $\mathcal{Z}$ is the feature space and $\mathcal{V}$ is the set of domain labels. Let $h_{\mathbf{v}}(\mathbf{z})$ be a conditional probability of $V$ where $\mathbf{v} \in \mathcal{V}$ given $Z = \mathbf{z}$, defined by $h$. Then the following holds:*

$$I(Z; V) = \max_{h_{\mathbf{v}}(\mathbf{z}):\sum_{\mathbf{v} \in \mathcal{V}} h_{\mathbf{v}}(\mathbf{z})=1, \forall \mathbf{z}} \sum_{\mathbf{v} \in \mathcal{V}} P_V(\mathbf{v}) \mathbb{E}_{\mathbf{z} \sim P_{Z|\mathbf{v}}} \big[ \log h_{\mathbf{v}}(\mathbf{z}) \big] + H(V) \tag{4}$$

The detailed proof is provided in the Roh et al. (2020) and Supplementary Material. As done in Roh et al. (2020), we can derive the empirical version of Theorem 2 as follows:

$$\hat{I}(Z; V) = \max_{h_{\mathbf{v}}(\mathbf{z}):\sum_{\mathbf{v} \in \mathcal{V}} h_{\mathbf{v}}(\mathbf{z})=1, \forall \mathbf{z}} \frac{1}{M} \sum_{\mathbf{v} \in \mathcal{V}} \sum_{i:\mathbf{v}_i=\mathbf{v}} \log h_{\mathbf{v}_i}(\mathbf{z}_i) + H(V), \tag{5}$$

where $M$ is the number of total representation samples, $i$ is the sample index, and $\mathbf{v}_i$ is the corresponding domain label of the $i$th sample. Using this equation, we combine our information-constrained objective function and the results of Lemma 1. For binary classification $\mathcal{V} = \{0, 1\}$ with $Z_S$ and $Z_T$ of equal size $M/2$, we propose the following information-regularized minimax problem:

$$\begin{aligned} F^*, C^* &= \arg\min_{F,C} \ L(F, C) + \beta \hat{I}(Z; V) \\ &= \arg\min_{F,C} \max_{h \in \mathcal{H}} \ L(F, C) + \beta \frac{1}{M} \big[ \sum_{i:\mathbf{z}_i \in Z_S} \log h(\mathbf{z}_i) + \sum_{j:\mathbf{z}_j \in Z_T} \log(1 - h(\mathbf{z}_j)) \big], \end{aligned} \tag{6}$$

where $\beta$ is a Lagrangian multiplier, $h(\mathbf{z}_i) \triangleq h_{\mathbf{v}_i=1}(\mathbf{z}_i)$ and $1 - h(\mathbf{z}_i) \triangleq h_{\mathbf{v}_i=0}(\mathbf{z}_i)$, with $h(\mathbf{z}_i)$ representing the probability that $\mathbf{z}_i$ belongs to the source domain. This setting automatically dismisses the condition $\sum_{\mathbf{v} \in \mathcal{V}} h_{\mathbf{v}}(\mathbf{z}) = 1, \forall \mathbf{z}$. Note that we have accommodated a simple situation in which the entropy $H(V)$ remains constant.

## 3.3 Advantages over other MDA methods

The relationship between (3) and (6) provides us a theoretical insights that the problem of minimizing mutual information between the latent representation and the domain label is closely related to minimizing the $\mathcal{H}$-divergence using the adversarial learning scheme. This relationship clearly underlines the significance of information regularization for MDA. Compared to the existing MDA approaches (Xu et al. (2018); Zhao et al. (2018)), which inevitably distribute domain-discriminative knowledge over $N$ different domain classifiers, the above objective function (6) enables us to seamlessly integrate such information with the single-domain classifier $h$.

Using a single domain discriminator also helps reduce the variance of gradient. Large variances in the stochastic gradients slow down the convergence, which leads to poor performance (Johnson & Zhang (2013)). Herein, we analyze the variances of the stochastic gradients of existing optimization constraints. By excluding the weighted source combination strategy, we can simplify the optimization constraint of existing adversarial MDA methods as sum of the information constraints:

$$\sum_{k=1}^{N} I(Z_k; U_k) = \sum_{k=1}^{N} \max_{h_{\mathbf{u}}^k(\mathbf{z}):\sum_{\mathbf{u} \in \mathcal{U}} h_{\mathbf{u}}^k(\mathbf{z})=1, \forall \mathbf{z}} \sum_{\mathbf{u} \in \mathcal{U}} P_{U_k}(\mathbf{u}) \mathbb{E}_{\mathbf{z}_k \sim P_{Z_k|\mathbf{u}}} \big[ \log h_{\mathbf{u}}^k(\mathbf{z}_k) \big] + \sum_{k=1}^{N} H(U_k), \tag{7}$$

where $U_k$ is the $k$th domain label with $\mathcal{U} = \{0, 1\}$, $P_{Z_k|\mathbf{u}=0}(\cdot) = P_{Z|\mathbf{v}=N+1}(\cdot)$ corresponding to the target domain, $P_{Z_k|\mathbf{u}=1}(\cdot) = P_{Z|\mathbf{v}=k}(\cdot)$ corresponding to the $k$th source domain, and $h_{\mathbf{u}}^k(\mathbf{z}_k)$ being the conditional probability of $\mathbf{u} \in \mathcal{U}$ given $\mathbf{z}_k$ defined by the $k$th discriminator indicating that the sample is generated from the $k$th source domain. Again, we treat the entropy $H(U_k)$ as a constant. Note that the interaction information cannot be measured with (7).

Given $M = m(N+1)$ samples with $m$ representing the number of samples per domain, an empirical version of (7) is:

$$\sum_{k=1}^{N} \hat{I}(Z_k; U_k) = \frac{1}{M} \sum_{k=1}^{N} \max_{h_{\mathbf{u}}^k(\mathbf{z}):\sum_{\mathbf{u} \in \mathcal{U}} h_{\mathbf{u}}^k(\mathbf{z})=1, \forall \mathbf{z}} \sum_{\mathbf{u} \in \mathcal{U}} \sum_{i:\mathbf{u}^i=\mathbf{u}} \log h_{\mathbf{u}}^k(\mathbf{z}_k^i) + \sum_{k=1}^{N} H(U_k). \tag{8}$$

Let $I_k$ be a shorthand for the $k$th term inside the first summation. Without loss of generality we make simplifying assumptions that all $Var[I_k]$ are the same for all k and so are $Cov[I_k, I_j]$ for all pairs. Then the variance of (8) is given by:

$$Var\Big[\sum_{k=1}^{N} \hat{I}(Z_k; U_k)\Big] = \frac{1}{M^2}\Big(\sum_{k=1}^{N} Var[I_k] + 2\sum_{k=1}^{N}\sum_{j=k}^{N} Cov[I_k, I_j]\Big) \tag{9}$$

$$= \frac{1}{m^2}\Big(\frac{N}{(N+1)^2} Var[I_k] + \frac{N(N-1)}{(N+1)^2} Cov[I_k, I_j]\Big).$$

As earlier works solve $N$ adversarial minimax problems, the covariance term is additionally included and its contribution to the variance does not decrease with increasing $N$. In other words, the covariance term may dominate the variance of the gradients as the number of domain increases. In contrast, the variance of our constraint (5) is inversely proportional to $(N+1)^2$. Let $I_m$ be a shorthand for the maximization term except $\frac{1}{M}$ in (5). Then the variance of (5) is given by:

$$Var\Big[\hat{I}(Z; V)\Big] = \frac{1}{m^2(N+1)^2}\Big(Var[I_m]\Big). \tag{10}$$

It implies that our framework can significantly improve the stability of stochastic gradient optimization compared to existing approaches, especially when the model is deemed to learn from many domains.

### 3.4 Situating domain adaptation in context of information bottleneck theory

In this Section, we bridge the gap between the existing adversarial DA method and the information bottleneck (IB) theory (Tishby et al. (2000); Tishby & Zaslavsky (2015); Alemi et al. (2016)). Tishby et al. (2000) examined the problem of learning an encoding $Z$ such that it is maximally informative about the class $Y$ while being minimally informative about the sample $X$:

$$\min_{P_{enc}(\mathbf{z}|\mathbf{x})} \beta I(Z; X) - I(Z; Y), \tag{11}$$

where $\beta$ is a Lagrangian multiplier. Indeed, the role of the bottleneck term $I(Z; X)$ matches our mutual information $I(Z; V)$ between the latent representation and the domain label. We foster close collaboration between two information bottleneck terms by incorporating those into $I(Z; X, V)$.

**Theorem 3.** *Let $P_{Z|\mathbf{x}, \mathbf{v}}(\mathbf{z})$ be a conditional probabilistic distribution of Z where $\mathbf{z} \in \mathcal{Z}$, defined by the encoder F, given a sample $\mathbf{x} \in \mathcal{X}$ and the domain label $\mathbf{v} \in \mathcal{V}$. Let $R_Z(\mathbf{z})$ denotes a prior marginal distribution of Z. Then the following inequality holds:*

$$I(Z; X, V) \leq \max_{h_{\mathbf{v}}(\mathbf{z}): \sum_{\mathbf{v} \in \mathcal{V}} h_{\mathbf{v}}(\mathbf{z})=1, \forall \mathbf{z}} \sum_{\mathbf{v} \in \mathcal{V}} P_V(\mathbf{v}) \mathbb{E}_{P_{\mathbf{z} \sim Z|\mathbf{v}}}\big[\log h_{\mathbf{v}}(\mathbf{z})\big] + H(V)$$
$$+ \mathbb{E}_{\mathbf{x}, \mathbf{v} \sim P_{X, V}}\big[D_{KL}[P_{Z|\mathbf{x}, \mathbf{v}} \parallel R_Z]\big] \tag{12}$$

The proof of Theorem 3 uses the chain rule: $I(Z; X, V) = I(Z; V) + I(Z; X \mid V)$. The detailed proof is provided in the Supplementary Material. Whereas the role of $I(Z; X \mid V)$ is to purify the latent representation generated from the given domain, $I(Z; V)$ serves as a proxy for regularization that aligns the purified representations across different domains. Thus, the existing DA approaches (Luo et al. (2019); Song et al. (2019)) using variational information bottleneck (Alemi et al. (2016)) can be reviewed as special cases for Theorem 3 with a single-source domain.

## 4 Multi-source Information-regularized Adaptation Networks

In this Section, we provide the details of our proposed architecture, referred to as a multi-source information-regularized adaptation network (**MIAN**). **MIAN** addresses the information-constrained min–max problem for MDA (Section 3.2) using the three subcomponents depicted in Figure 1: information regularization, source classification, and Decaying Batch Spectral Penalization (DBSP).

**Information regularization.** To estimate the empirical mutual information $\hat{I}(Z; V)$ in (5), the domain classifier $h$ should be trained to minimize softmax cross enropy. Let $\mathcal{V} = \{1, 2, ..., N+1\}$

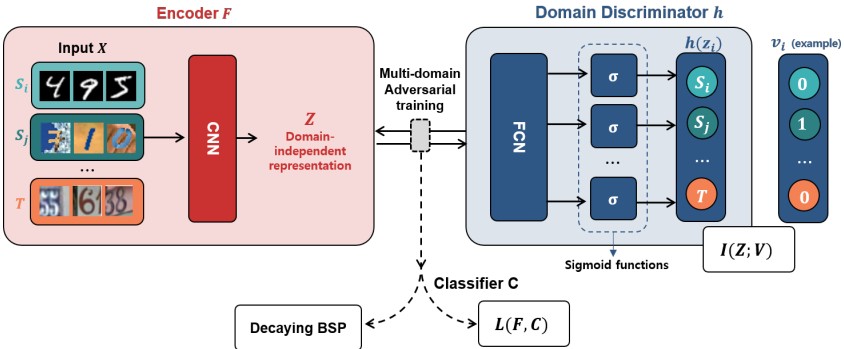

Figure 1: Proposed neural architecture for multi-source domain adaptation: Multi-source Information-regularized Adaptation Network (MIAN). Multi-source and target domain input data are fed into the encoder. We denote arbitrary source domains as $S_i$ and $S_j$ without loss of generality. The domain discriminator outputs a logit vector, where each dimension corresponds to each domain. CNN and FCN refers to convolutional neural networks and fully connected neural networks, respectively.

and denote $h(\mathbf{z})$ as $N+1$ dimensional vector of the conditional probability for each domain given the sample $\mathbf{z}$. Let $\mathbb{1}$ be a $N+1$ dimensional vector of all ones, and $\mathbb{1}_{[k=\mathbf{v}]}$ be a $N+1$ dimensional vector whose $\mathbf{v}$th value is 1 and 0 otherwise. Given $M = m(N+1)$ samples, the objective is:

$$\min_h -\frac{1}{M} \sum_{\mathbf{v} \in \mathcal{V}} \sum_{i:\mathbf{v}_i = \mathbf{v}} \left[ \mathbb{1}_{[k=\mathbf{v}_i]}^T \log h(\mathbf{z}_i) \right]. \tag{13}$$

In this study, we slightly modify the objective (13). Specifically, we explicitly minimized the conditional probability of the remaining domains excepting the $\mathbf{v}$th domain. Let $\mathbb{1}_{[k \neq \mathbf{v}]}$ be the flipped version of $\mathbb{1}_{[k=\mathbf{v}]}$. Then the objective function for the domain discriminator is:

$$\min_h -\frac{1}{M} \sum_{\mathbf{v} \in \mathcal{V}} \sum_{i:\mathbf{v}_i = \mathbf{v}} \left[ \mathbb{1}_{[k=\mathbf{v}_i]}^T \log h(\mathbf{z}_i) + \mathbb{1}_{[k \neq \mathbf{v}_i]}^T \log(\mathbb{1} - h(\mathbf{z}_i)) \right], \tag{14}$$

where the objective function for encoder training is to maximize (14). Our objective function is also closely related to that of GAN (Goodfellow et al. (2014)), and we experimentally found that using the variant objective function of GAN (Mao et al. (2017)) works slightly better.

The above objective is closely related to optimizing every pairwise domain discrepancy between the given domain and the mixture of the others. Let each $D_\mathbf{v}$ and $D_{\mathbf{v}^c}$ represent the $\mathbf{v}$th domain and the mixture of the remaining $N$ domains with the same mixture weight $\frac{1}{N}$, relatively. Then we can define $\mathcal{H}$-divergence as $d_\mathcal{H}(D_\mathbf{v}, D_{\mathbf{v}^c})$, and an average of such $\mathcal{H}$-divergence for every $\mathbf{v}$ as $d_\mathcal{H}(\mathcal{V})$. Assume that the samples of size $m$, $Z_\mathbf{v}$ and $Z_{\mathbf{v}^c}$, are generated from each $D_\mathbf{v}$ and $D_{\mathbf{v}^c}$, where $Z_{\mathbf{v}^c} = \bigcup_{\mathbf{v}' \neq \mathbf{v}} Z_{\mathbf{v}'}$ with $|Z_{\mathbf{v}'}| = m/N$ for all $\mathbf{v}' \in \mathcal{V}$. Thus the domain label $\mathbf{v}_j \neq \mathbf{v}$ for every $j$th sample in $Z_{\mathbf{v}^c}$. Then the average of empirical $\mathcal{H}$-divergence $\hat{d}_\mathcal{H}(\mathcal{V})$ is defined as follows:

$$\hat{d}_\mathcal{H}(\mathcal{V}) = \frac{1}{N+1} \sum_{\mathbf{v} \in \mathcal{V}} \hat{d}_\mathcal{H}(Z_\mathbf{v}, Z_{\mathbf{v}^c})$$
$$= \frac{1}{N+1} \sum_{\mathbf{v} \in \mathcal{V}} 2\left(1 - \min_{h \in \mathcal{H}} \left[ \frac{1}{m} \sum_{i:\mathbf{v}_i = \mathbf{v}} \mathbb{I}[h_\mathbf{v}(\mathbf{z}_i) = 1] + \frac{1}{m} \sum_{j:\mathbf{v}_j \neq \mathbf{v}} \mathbb{I}[h_\mathbf{v}(\mathbf{z}_j) = 0] \right] \right), \tag{15}$$

where $h_\mathbf{v}(\mathbf{z})$ represents the $\mathbf{v}$th value of $h(\mathbf{z})$. Note that $h(\mathbf{z})$ corresponds to $N+1$ dimensional one-hot classification vector in (15), unlike in (14). Then, let $\mathbf{I}[h(\mathbf{z})] := \left[ \mathbb{I}(h_\mathbf{v}(\mathbf{z}) = 1) \right]_{\mathbf{v} \in \mathcal{V}}$ be the $N+1$ dimensional one-hot indicator vector. Given the unified domain discriminator $h$ in the inner minimization for every $\mathbf{v}$ in (15), we train $h$ to approximate the lower bound of $\hat{d}_\mathcal{H}(\mathcal{V})$ as follows:

$$h^* = \arg\max_{h \in \mathcal{H}} \frac{1}{M} \sum_{\mathbf{v} \in \mathcal{V}} \left( \sum_{i:\mathbf{v}_i = \mathbf{v}} \mathbb{I}[h_\mathbf{v}(\mathbf{z}_i) = 1] + \sum_{j:\mathbf{v}_j \neq \mathbf{v}} \mathbb{I}[h_\mathbf{v}(\mathbf{z}_j) = 0] \right)$$
$$= \arg\min_{h \in \mathcal{H}} -\frac{1}{M} \sum_{\mathbf{v} \in \mathcal{V}} \sum_{i:\mathbf{v}_i = \mathbf{v}} \mathbb{1}_{[k=\mathbf{v}_i]}^T \mathbf{I}[h(\mathbf{z}_i)] + \mathbb{1}_{[k \neq \mathbf{v}_i]}^T \left( \mathbb{1} - \mathbf{I}[h(\mathbf{z}_i)] \right), \tag{16}$$

where the latter equality is obtained by rearranging the summation terms in the first equality.

Based on the close relationship between (14) and (16), we can make the link between information regularization and $\mathcal{H}$-divergence optimization given multi-source domain; minimizing $\hat{d}_{\mathcal{H}}(\mathcal{V})$ is closely related to implicit regularization of the mutual information between latent representations and domain labels. Because the output vector $h(\mathbf{z})$ in (15) often comes from the $\arg\max$ operation, (15) is not differentiable w.r.t. $\mathbf{z}$. However, our framework has a differentiable objective as in (14).

There are two benefits of minimizing $d_{\mathcal{H}}(\mathcal{V})$. First, it includes $\mathcal{H}$-divergence between the target and a mixture of sources, which directly affects the upper bound of the empirical risk on target samples (Theorem 5 in Ben-David et al. (2010)). Second, $d_{\mathcal{H}}(\mathcal{V})$ lower-bounds the average of every pairwise $\mathcal{H}$-divergence between domains. The detailed proof is provided in the appendix (Lemma 2). Note that unlike our single domain classifier setting, existing methods (Li et al. (2018)) require a number of about $\mathcal{O}(N^2)$ domain classifiers to approximate all pairwise combinations of domain discrepancy.

**Source classification.** Along with learning domain-independent latent representations illustrated in the above, we train the classifier with the labeled source domain datasets that can be directly applied to the target domain representations in practice. To minimize the empirical risk on source domain, we use a generic softmax cross-entropy loss function with labeled source domain samples as $L(F, C)$.

**Decaying batch spectral penalization.** Applying above information-theoretic insights, we further describe a potential side effect of existing adversarial DA methods. Information regularization may lead to overriding implicit entropy minimization, particularly in the early stages of the training, impairing the richness of latent feature representations. To prevent such a pathological phenomenon, we introduce a new technique called Decaying Batch Spectral Penalization (DBSP), which is intended to control the SVD entropy of the feature space. Our version improves training efficiency compared to original Batch Spectral Penalization (Chen et al. (2019)). We refer to this version of our model as **MIAN-$\gamma$**. As vanila **MIAN** is sufficient to outperform other state-of-the-art methods (Section 5), **MIAN-$\gamma$** is further discussed in the Supplementary Material.

## 5 EXPERIMENTS

Table 1: Accuracy (%) on Digits-Five dataset. SYNTH denotes Synthetic Digits (Ganin & Lempitsky (2014)). The baseline results for the Digits-Five dataset were taken from (Peng et al. (2019)).

| Standards | Models | MNIST-M | MNIST | USPS | SVHN | SYNTH | Avg |
|---|---|---|---|---|---|---|---|
| Source-combined | Source Only | 63.70 | 92.30 | 90.71 | 71.51 | 83.44 | 80.33 |
| | DAN | 67.87 | 97.50 | 93.49 | 67.80 | 86.93 | 82.72 |
| | DANN | 70.81 | 97.90 | 93.47 | 68.50 | 87.37 | 83.61 |
| Single-best | Source Only | 63.37 | 90.50 | 88.71 | 63.54 | 82.44 | 77.71 |
| | DAN | 63.78 | 96.31 | 94.24 | 62.45 | 85.43 | 80.44 |
| | CORAL | 62.53 | 97.21 | 93.45 | 64.40 | 82.77 | 80.07 |
| | DANN | 71.30 | 97.60 | 92.33 | 63.48 | 85.34 | 82.01 |
| | JAN | 65.88 | 97.21 | 95.42 | 75.27 | 86.55 | 84.07 |
| | ADDA | 71.57 | 97.89 | 92.83 | 75.48 | 86.45 | 84.84 |
| | MEDA | 71.31 | 96.47 | 97.01 | 78.45 | 84.62 | 85.60 |
| | MCD | 72.50 | 96.21 | 95.33 | 78.89 | 87.47 | 86.10 |
| Multi-source | DCTN | 70.53 | 96.23 | 92.81 | 77.61 | 86.77 | 84.79 |
| | M³SDA | 69.76 | **98.58** | 95.23 | 78.56 | 87.56 | 86.13 |
| | M³SDA-$\beta$ | 72.82 | 98.43 | 96.14 | 81.32 | 89.58 | 87.65 |
| | **MIAN** | **84.36** | 97.91 | **96.49** | **88.18** | **93.23** | **92.03** |

To assess the performance of **MIAN**, we ran a large-scale simulation using the following benchmark datasets: Digits-Five, Office-31 and Office-Home. For a fair comparison, we reproduced all the other baseline results using the same backbone architecture and optimizer settings as the proposed method. For the source-only and single-source DA standards, we introduce two MDA approaches (Xu et al. (2018); Peng et al. (2019)): (1) source-combined, i.e., all source-domains are incorporated into a

single source domain; (2) single-best, i.e., the best adaptation performance on the target domain is reported. Owing to limited space, details about simulation settings, used baseline models and datasets are presented in the Supplementary Material.

Table 2: Accuracy (%) on Office-31 dataset.

| Standards | Models | Amazon | DSLR | Webcam | Avg |
|---|---|---|---|---|---|
| Single-best | Source Only | 55.23±0.72 | 95.59±1.37 | 87.06±1.50 | 79.29 |
| | DAN | 64.19±0.56 | **100.00±0.00** | 97.45±0.44 | 87.21 |
| | JAN | 69.57±0.27 | 99.80±0.00 | 97.4±0.26 | 88.92 |
| Source-combined | Source Only | 60.80±2.00 | 92.68±0.31 | 86.91±2.37 | 80.13 |
| | JAN | 70.15±0.19 | 95.20±0.36 | 95.15±0.23 | 86.83 |
| | DANN | 68.15±0.42 | 97.59±0.60 | 96.77±0.26 | 87.50 |
| | DAN | 65.77±0.74 | 99.26±0.23 | 97.51±0.41 | 87.51 |
| | DANN+BSP | 71.13±0.44 | 96.65±0.30 | 98.32±0.26 | 88.70 |
| | MCD | 68.57±1.06 | 99.49±0.25 | **99.30±0.38** | 89.12 |
| Multi-source | DCTN | 62.74±0.50 | 99.44±0.25 | 97.92±0.29 | 86.70 |
| | M³SDA | 67.19±0.22 | 99.34±0.19 | 98.04±0.21 | 88.19 |
| | M³SDA-$\beta$ | 69.41±0.82 | 99.64±0.19 | **99.30±0.31** | 89.45 |
| | **MIAN** | **74.65±0.48** | 99.48±0.35 | 98.49±0.59 | **90.87** |

Table 3: Accuracy (%) on Office-Home dataset.

| Standards | Models | Art | Clipart | Product | Realworld | Avg |
|---|---|---|---|---|---|---|
| Source-combined | Source Only | 64.58±0.68 | 52.32±0.63 | 77.63±0.23 | 80.70±0.81 | 68.81 |
| | DANN | 64.26±0.59 | 58.01±1.55 | 76.44±0.47 | 78.80±0.49 | 69.38 |
| | DANN+BSP | 66.10±0.27 | 61.03±0.39 | 78.13±0.31 | 79.92±0.13 | 71.29 |
| | DAN | 68.28±0.45 | 57.92±0.65 | 78.45±0.05 | **81.93±0.35** | 71.64 |
| | MCD | 67.84±0.38 | 59.91±0.55 | 79.21±0.61 | 80.93±0.18 | 71.97 |
| Multi-source | M³SDA | 66.22±0.52 | 58.55±0.62 | 79.45±0.52 | 81.35±0.19 | 71.39 |
| | DCTN | 66.92±0.60 | 61.82±0.46 | 79.20±0.58 | 77.78±0.59 | 71.43 |
| | **MIAN** | **69.39±0.50** | **63.05±0.61** | **79.62±0.16** | 80.44±0.24 | **73.12** |

## 5.1 SIMULATION RESULTS

The classification accuracy for Digits-Five, Office-31, and Office-Home are summarized in Tables 1, 2, and 3, respectively. We found that **MIAN** outperforms most of other state-of-the-art single-source and multi-source DA methods by a large margin. Note that our method demonstrated a significant improvement in challenging domains, such as MNIST-M, Amazon or Clipart.

## 5.2 QUALITATIVE AND QUANTITATIVE ANALYSES

**Design of domain discriminator.** To quantify the extent to which performance improvement is achieved by unifying the domain discriminators, we compared the performances of the four different versions of **MIAN** (Figure 2a, 2b). *No S-S align* is the same as **MIAN** with the exception that only the target and each source domains are aligned. *No LS* uses the objective function as in (14), and unlike (Mao et al. (2017)). *Multi D* employs as many discriminators as the number of source domains which is analogous to the existing approaches. For a fair comparison, all the other experimental settings are fixed. The results illustrate that all the versions with the unified discriminator reliably outperform *Multi D* in terms of both accuracy and reliability. This suggests that unification of the domain discriminators can substantially improves the task performance.

**Variance of stochastic gradients.** With respect to the above analysis, we compared the variance of the stochastic gradients computed with different available domain discriminators. We trained **MIAN** and *Multi D* using mini-batches of samples. The number of samples in a batch was fixed as 128 per

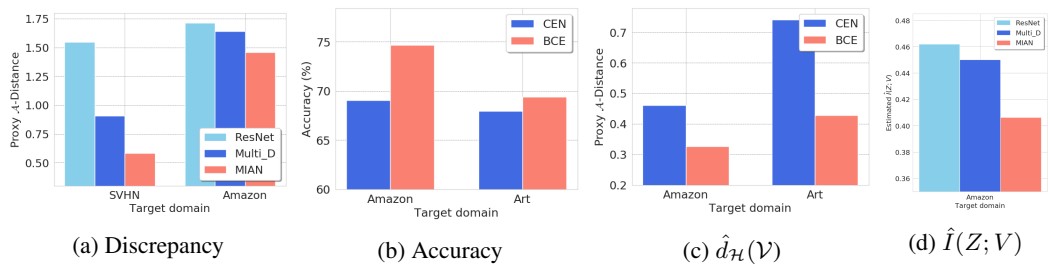

(a)        (b)        (c)        (d)

Figure 2: **(a)**∼**(b)**: Test accuracies for **(a)** MNIST-M and **(b)** SVHN as target domains. **(c)**∼**(d)**: Variance of stochastic gradients after 1000 steps for **(c)** MNIST-M and **(d)** SVHN as target domains in log scale. Less is better.

(a) Discrepancy       (b) Accuracy       (c) $\hat{d}_{\mathcal{H}}(\mathcal{V})$       (d) $\hat{I}(Z;V)$

Figure 3: **(a)** Proxy $\mathcal{A}$-distance. **(b)**∼**(c)** Ablation study on the objective of domain discriminator. *CEN* stands for multi-class cross entropy loss in (13), while BCE stands for binary-class cross entropy losses in (14). **(d)** Empirical information $\hat{I}(Z;V)$. We treat $H(V) = \log|\mathcal{V}|$.

domain. After the early stages of training, we computed the gradients for the weights and biases of both the top and bottom layers of the encoder on the full training set. Figures 2c, 2d show that **MIAN** with the unified discriminator yields exponentially lower variance of the gradients compared to *Multi D*. Thus it is more feasible to use the unified discriminator when a large number of domains are given.

**Proxy $\mathcal{A}$-distance.** To analyze the performance improvement in depth, we measured Proxy $\mathcal{A}$-Distance (PAD) as an empirical approximation of domain discrepancy (Ganin et al. (2016)). Given the generalization error $\epsilon$ on discriminating between the target and source samples, PAD is defined as $\hat{d}_{\mathcal{A}} = 2(1-2\epsilon)$. Figure 3a shows that **MIAN** yields lower PAD between the source and target domain on average, potentially associated with optimizing $\hat{d}_{\mathcal{H}}(\mathcal{V})$. To test this conjecture, we conducted an ablation study on the objective of domain discriminator (Figure 3b, 3c). All the other experimental settings were fixed except for using the objective of the unified domain discriminator as (13), or (14). While both cases help the adaptation, using (14) yields lower $\hat{d}_{\mathcal{H}}(\mathcal{V})$ and higher test accuracy.

**Estimation of mutual information.** We measure the empirical mutual information $\hat{I}(Z;V)$ with assuming $H(V)$ as a constant. For the measurement, we trained the domain discriminator to minimize the softmax cross entropy (13) with sufficient iterations. Figure 3d shows that **MIAN** yields the lowest $\hat{I}(Z;V)$, guaranteeing that the obtained representation achieves low-level of domain dependence.

## 6   CONCLUSION

In this paper, we have presented a unified information-regularization framework for MDA. The proposed framework allows us to examine the existing adversarial DA methods and also motivates us to implement a novel neural architecture for MDA. We provided both theoretical arguments and empirical evidence to fully justify three potential pitfalls of using multiple discriminators: dispersed domain-discriminative knowledge, lack of scalability and high variance in the objective. Our framework also establishes a bridge between adversarial DA and Information Bottleneck theory. The proposed model does not require complicated settings such as image generation, pretraining, multiple discriminators, multiple encoders or classifiers, which are often adopted in the existing MDA methods (Zhao et al. (2019b;c); Wang et al. (2019)).

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

## A PROOFS

In this Section, we present the detailed proofs for Theorems 2 and 3, explained in the main paper. We also present Lemma 2, as mentioned in Section 4. Following (Roh et al. (2020)), we provide a proof of Theorem 2 below for the sake of completeness.

### A.1 PROOF OF THEOREM 2

**Theorem 2.** *Let $P_Z(\mathbf{z})$ be the distribution of $Z$ where $\mathbf{z} \in \mathcal{Z}$. Let $h$ be a domain classifier $h : \mathcal{Z} \to \mathcal{V}$, where $\mathcal{Z}$ is the feature space and $\mathcal{V}$ is the set of domain labels. Let $h_{\mathbf{v}}(Z)$ be a conditional probability of $V$ where $\mathbf{v} \in \mathcal{V}$ given $Z = \mathbf{z}$, defined by $h$. Then the following holds:*

$$I(Z;V) = \max_{h_{\mathbf{v}}(\mathbf{z}): \sum_{\mathbf{v} \in \mathcal{V}} h_{\mathbf{v}}(\mathbf{z})=1, \forall \mathbf{z}} \sum_{\mathbf{v} \in \mathcal{V}} P_V(\mathbf{v}) \mathbb{E}_{\mathbf{z} \sim P_{Z|\mathbf{v}}} \big[ \log h_{\mathbf{v}}(\mathbf{z}) \big] + H(V) \tag{17}$$

*Proof.* By definition,

$$\begin{aligned} I(Z;V) &= D_{KL}\big(P(Z,V) \parallel P(Z)P(V)\big) \\ &= \sum_{\mathbf{v} \in \mathcal{V}} P_V(\mathbf{v}) \mathbb{E}_{\mathbf{z} \sim P_{Z|\mathbf{v}}} \Big[ \log \frac{P_{Z,V}(\mathbf{z}, \mathbf{v})}{P_Z(\mathbf{z})} \Big] + H(V) \end{aligned} \tag{18}$$

Let us constrain the term inside the log by $h_{\mathbf{v}}(\mathbf{z}) = \frac{P_{Z,V}(\mathbf{z},\mathbf{v})}{P_Z(\mathbf{z})}$ where $h_{\mathbf{v}}(\mathbf{z})$ represents the conditional probability of $V = \mathbf{v}$ for any $\mathbf{v} \in \mathcal{V}$ given $Z = \mathbf{z}$. Then we have: $\sum_{\mathbf{v} \in \mathcal{V}} h_{\mathbf{v}}(\mathbf{z}) = 1$ for all possible values of $\mathbf{z}$ according to the law of total probability. Let $\mathbf{h}$ denote the collection of $h_{\mathbf{v}}(\mathbf{z})$ for all possible values of $\mathbf{v}$ and $\mathbf{z}$, and $\boldsymbol{\lambda}$ be the collection of $\lambda_{\mathbf{z}}$ for all values of $\mathbf{z}$. Then, we can construct the Lagrangian function by incorporating the constraint $\sum_{\mathbf{v} \in \mathcal{V}} h_{\mathbf{v}}(\mathbf{z}) = 1$ as follows:

$$L(\mathbf{h}, \boldsymbol{\lambda}) = \sum_{\mathbf{v} \in \mathcal{V}} P_V(\mathbf{v}) \mathbb{E}_{\mathbf{z} \sim P_{Z|\mathbf{v}}} \Big[ log\big(h_{\mathbf{v}}(\mathbf{z})\big) \Big] + H(V) + \sum_{\mathbf{z} \in \mathcal{Z}} \lambda_{\mathbf{z}} \Big(1 - \sum_{\mathbf{v} \in \mathcal{V}} h_{\mathbf{v}}(\mathbf{z})\Big) \tag{19}$$

We can use the following KKT conditions:

$$\frac{\partial L(\mathbf{h}, \boldsymbol{\lambda})}{\partial h_{\mathbf{v}}(\mathbf{z})} = P_V(\mathbf{v}) \frac{P_{Z|\mathbf{v}}(\mathbf{z})}{h_{\mathbf{v}}^*(\mathbf{z})} - \lambda_{\mathbf{z}}^* = 0, \quad \forall (\mathbf{z}, \mathbf{v}) \in \mathcal{Z} \times \mathcal{V} \tag{20}$$

$$1 - \sum_{\mathbf{v} \in \mathcal{V}} h_{\mathbf{v}}^*(\mathbf{z}) = 0, \quad \forall \mathbf{z} \in \mathcal{Z} \tag{21}$$

Solving the two equations, we have $1 - \sum_{\mathbf{v} \in \mathcal{V}} \frac{P_V(\mathbf{v})P_{Z|\mathbf{v}}(\mathbf{z})}{\lambda_{\mathbf{z}}^*} = 0$ such that $\lambda_{\mathbf{z}}^* = P_Z(\mathbf{z})$ for all $\mathbf{z}$. Then for all the possible values of $\mathbf{z}$,

$$\begin{aligned} h_{\mathbf{v}}^*(\mathbf{z}) &= \frac{P_{Z,V}(\mathbf{z}, \mathbf{v})}{P_Z(\mathbf{z})} \\ &= P_{V|\mathbf{z}}(\mathbf{v}), \end{aligned} \tag{22}$$

where the given $h_{\mathbf{v}}^*(\mathbf{z})$ is same as the term inside log in (18). Thus, the optimal solution of concave Lagrangian function (19) obtained by $h_{\mathbf{v}}^*(\mathbf{z})$ is equal to the mutual information in (18). The substitution of $h_{\mathbf{v}}^*(\mathbf{z})$ into (18) completes the proof. □

Our framework can further be applied to segmentation problems because it provides a new perspective on pixel space (Sankaranarayanan et al. (2018a;b); Murez et al. (2018)) and segmentation space (Tsai et al. (2018)) adaptation. The generator in pixel space and segmentation space adaptation learns to transform images or segmentation results from one domain to another. In the context of information regularization, we can view these approaches as limiting information $I(\hat{X}; V)$ between the generated output $\hat{X}$ and the domain label $V$, which is accomplished by involving the encoder for pixel-level generation. This alleviates the domain shift in a raw pixel level. Note that one can choose between limiting the feature-level or pixel-level mutual information. These different regularization terms may be complementary to each other depending on the given task.

## A.2  PROOF OF THEOREM 3

**Theorem 3.** *Let $P_{Z|\mathbf{x},\mathbf{v}}(\mathbf{z})$ be a conditional probabilistic distribution of $Z$ where $\mathbf{z} \in \mathcal{Z}$, defined by the encoder $F$, given a sample $\mathbf{x} \in \mathcal{X}$ and the domain label $\mathbf{v} \in \mathcal{V}$. Let $R_Z(\mathbf{z})$ denotes a prior marginal distribution of $Z$. Then the following inequality holds:*

$$I(Z; X, V) \leq \max_{h_\mathbf{v}(\mathbf{z}):\sum_{\mathbf{v}\in\mathcal{V}} h_\mathbf{v}(\mathbf{z})=1, \forall \mathbf{z}} \sum_{\mathbf{v}\in\mathcal{V}} P_V(\mathbf{v})\mathbb{E}_{P_{\mathbf{z}\sim Z|\mathbf{v}}}\big[\log h_\mathbf{v}(\mathbf{z})\big] + H(V)$$
$$+ \mathbb{E}_{\mathbf{x},\mathbf{v}\sim P_{X,V}}\big[D_{KL}[P_{Z|\mathbf{x},\mathbf{v}} \parallel R_Z]\big] \tag{23}$$

*Proof.* Based on the chain rule for mutual information,

$$I(Z; X, V) = I(Z; V) + I(Z; X \mid V)$$
$$= \max_{h_\mathbf{v}(\mathbf{z}):\sum_{\mathbf{v}\in\mathcal{V}} h_\mathbf{v}(\mathbf{z})=1, \forall \mathbf{z}} \sum_{\mathbf{v}\in\mathcal{V}} P_V(\mathbf{v})\mathbb{E}_{\mathbf{z}\sim P_{Z|\mathbf{v}}}\big[\log h_\mathbf{v}(\mathbf{z})\big] + H(V) + I(Z; X \mid V), \tag{24}$$

where the latter equality is given by Theorem 2. Considering $I(Z; X \mid V)$,

$$I(Z; X \mid V) = \mathbb{E}_{\mathbf{v}\sim P_V}\Big[\mathbb{E}_{\mathbf{z},\mathbf{x}\sim P_{Z,X|\mathbf{v}}}\Big[\log \frac{P_{Z,X|\mathbf{v}}(\mathbf{z},\mathbf{x})}{P_{Z|\mathbf{v}}(\mathbf{z})P_{X|\mathbf{v}}(\mathbf{x})}\Big]\Big]$$
$$= \mathbb{E}_{\mathbf{x},\mathbf{v}\sim P_{X,V}}\Big[\mathbb{E}_{\mathbf{z}\sim P_{Z|\mathbf{x},\mathbf{v}}}\Big[\log \frac{P_{Z|\mathbf{x},\mathbf{v}}(\mathbf{z})}{P_{Z|\mathbf{v}}(\mathbf{z})}\Big]\Big]$$
$$= \mathbb{E}_{\mathbf{x},\mathbf{v}\sim P_{X,V}}\Big[\mathbb{E}_{\mathbf{z}\sim P_{Z|\mathbf{x},\mathbf{v}}}\big[\log P_{Z|\mathbf{x},\mathbf{v}}(\mathbf{z})\big]\Big] - \mathbb{E}_{\mathbf{v}\sim P_V}\Big[\mathbb{E}_{\mathbf{z}\sim P_{Z|\mathbf{v}}}\big[\log P_{Z|\mathbf{v}}(\mathbf{z})\big]\Big] \tag{25}$$
$$\leq \mathbb{E}_{\mathbf{x},\mathbf{v}\sim P_{X,V}}\Big[\mathbb{E}_{\mathbf{z}\sim P_{Z|\mathbf{x},\mathbf{v}}}\big[\log P_{Z|\mathbf{x},\mathbf{v}}(\mathbf{z})\big]\Big] - \mathbb{E}_{\mathbf{v}\sim P_V}\Big[\mathbb{E}_{\mathbf{z}\sim P_{Z|\mathbf{v}}}\big[\log R_Z(\mathbf{z})\big]\Big]$$
$$= \mathbb{E}_{\mathbf{x},\mathbf{v}\sim P_{X,V}}\Big[\mathbb{E}_{\mathbf{z}\sim P_{Z|\mathbf{x},\mathbf{v}}}\Big[\log \frac{P_{Z|\mathbf{x},\mathbf{v}}(\mathbf{z})}{R_Z(\mathbf{z})}\Big]\Big]$$
$$= \mathbb{E}_{\mathbf{x},\mathbf{v}\sim P_{X,V}}\big[D_{KL}\big[P_{Z|\mathbf{x},\mathbf{v}} \parallel R_Z\big]\big]$$

The second equality is obtained by using $P_{Z,X|\mathbf{v}}(\mathbf{z},\mathbf{x}) = P_{X|\mathbf{v}}(\mathbf{x})P_{Z|\mathbf{x},\mathbf{v}}(\mathbf{z})$. The inequality is obtained by using $D_{KL}[P_{Z|\mathbf{v}} \parallel R_Z] = \mathbb{E}_{\mathbf{z}\sim P_{Z|\mathbf{v}}}\big[\log P_{Z|\mathbf{v}}(\mathbf{z}) - \log R_Z(\mathbf{z})\big] \geq 0$, where $R_Z(\mathbf{z})$ is a variational approximation of the prior marginal distribution of $Z$. The last equality is obtained from the definition of KL-divergence. The substitution of (25) into (24) completes the proof. $\square$

The existing DA work on semantic segmentation tasks (Luo et al. (2019); Song et al. (2019)) can be explained as the process of fostering close collaboration between the aforementioned information bottleneck terms. The only difference between Theorem 3 for $\mathcal{V} = \{0, 1\}$ and the objective function in (Luo et al. (2019)) is that (Luo et al. (2019)) employed the shared encoding $P_{Z|\mathbf{x}}(\mathbf{z})$ instead of $P_{Z|\mathbf{x},\mathbf{v}}(\mathbf{z})$, whereas some adversarial DA approaches use the unshared one (Tzeng et al. (2017)).

## A.3  PROOF OF LEMMA 2

**Lemma 2.** *Let $d_\mathcal{H}(\mathcal{V}) = \frac{1}{N+1}\sum_{\mathbf{v}\in\mathcal{V}} d_\mathcal{H}(D_\mathbf{v}, D_{\mathbf{v}^c})$. Let $\mathcal{H}$ be a hypothesis class. Then,*

$$d_\mathcal{H}(\mathcal{V}) \leq \frac{1}{N(N+1)}\sum_{\mathbf{v},\mathbf{u}\in\mathcal{V}} d_\mathcal{H}(D_\mathbf{v}, D_\mathbf{u}) \tag{26}$$

*Proof.* Let $\alpha = \frac{1}{N}$ represents the uniform domain weight for the mixture of domain $D_{\mathbf{v}^c}$. Then,

$$
\begin{aligned}
d_{\mathcal{H}}(\mathcal{V}) &= \frac{1}{N+1} \sum_{\mathbf{v} \in \mathcal{V}} d_{\mathcal{H}}(D_{\mathbf{v}}, D_{\mathbf{v}^c}) \\
&= \frac{1}{N+1} \sum_{\mathbf{v} \in \mathcal{V}} 2 \sup_{h \in \mathcal{H}} \left| \mathbb{E}_{\mathbf{x} \sim P_{D_{\mathbf{v}}^X}} \left[ \mathbb{I}\big(h(\mathbf{x} = 1)\big) \right] - \mathbb{E}_{\mathbf{x} \sim P_{D_{\mathbf{v}^c}^X}} \left[ \mathbb{I}\big(h(\mathbf{x} = 1)\big) \right] \right| \\
&= \frac{1}{N+1} \sum_{\mathbf{v} \in \mathcal{V}} 2 \sup_{h \in \mathcal{H}} \left| \sum_{\mathbf{u} \in \mathcal{V}: \mathbf{u} \neq \mathbf{v}} \alpha \Big( \mathbb{E}_{\mathbf{x} \sim P_{D_{\mathbf{v}}^X}} \left[ \mathbb{I}\big(h(\mathbf{x} = 1)\big) \right] - \mathbb{E}_{\mathbf{x} \sim P_{D_{\mathbf{u}}^X}} \left[ \mathbb{I}\big(h(\mathbf{x} = 1)\big) \right] \Big) \right| \\
&\leq \frac{1}{N+1} \sum_{\mathbf{v} \in \mathcal{V}} \sum_{\mathbf{u} \in \mathcal{V}: \mathbf{u} \neq \mathbf{v}} \alpha \cdot 2 \sup_{h \in \mathcal{H}} \left| \mathbb{E}_{\mathbf{x} \sim P_{D_{\mathbf{v}}^X}} \left[ \mathbb{I}\big(h(\mathbf{x} = 1)\big) \right] - \mathbb{E}_{\mathbf{x} \sim P_{D_{\mathbf{u}}^X}} \left[ \mathbb{I}\big(h(\mathbf{x} = 1)\big) \right] \right| \\
&= \frac{1}{N(N+1)} \sum_{\mathbf{v}, \mathbf{u} \in \mathcal{V}} d_{\mathcal{H}}(D_{\mathbf{v}}, D_{\mathbf{u}}),
\end{aligned}
\tag{27}
$$

where the inequality follows from the triangluar inequality and jensen's inequality.

$\square$

## B  EXPERIMENTAL SETUP

In this Section, we describe the datasets, network architecture and hyperparameter configuration.

### B.1  DATASETS

We validate the Multi-source Information-regularized Adaptation Networks (**MIAN**) with the following benchmark datasets: Digits-Five, Office-31 and Office-Home. Every experiment is repeated four times and the average accuracy in target domain is reported.

**Digits-Five** (Peng et al. (2019)) dataset is a unified dataset including five different digit datasets: MNIST (LeCun et al. (1998)), MNIST-M (Ganin & Lempitsky (2014)), Synthetic Digits (Ganin & Lempitsky (2014)), SVHN, and USPS. Following the standard protocols of unsupervised MDA (Xu et al. (2018); Peng et al. (2019)), we used 25000 training images and 9000 test images sampled from a training and a testing subset for each of MNIST, MNIST-M, SVHN, and Synthetic Digits. For USPS, all the data is used owing to the small sample size. All the images are bilinearly interpolated to $32 \times 32$.

**Office-31** (Saenko et al. (2010)) is a popular benchmark dataset including 31 categories of objects in an office environment. Note that it is a more difficult problem than Digits-Five, which includes 4652 images in total from the three domains: Amazon, DSLR, and Webcam. All the images are interpolated to $224 \times 224$ using bicubic filters.

**Office-Home** (Venkateswara et al. (2017)) is a challenging dataset that includes 65 categories of objects in office and home environments. It includes 15,500 images in total from the four domains: Artistic images (Art), Clip Art(Clipart), Product images (Product), and Real-World images (Realworld). All the images are interpolated to $224 \times 224$ using bicubic filters.

### B.2  ARCHITECTURES

For the Digits-Five dataset, we use the same network architecture and optimizer setting as in (Peng et al. (2019)). For all the other experiments, the results are based on ResNet-50, which is pre-trained on ImageNet. The domain discriminator is implemented as a three-layer neural network. Detailed architecture is shown in Figure 5.

We compare our method with the following state-of-the-art domain adaptation methods: Deep Adaptation Network (**DAN**, Long et al. (2015)), Joint Adaptation Network (**JAN**, Long et al. (2017)), Manifold Embedded Distribution Alignment (**MEDA**, Wang et al. (2018)), Correlation Alignment (**CORAL**, Sun et al. (2016)), Domain Adversarial Neural Network (**DANN**, Ganin et al. (2016)),

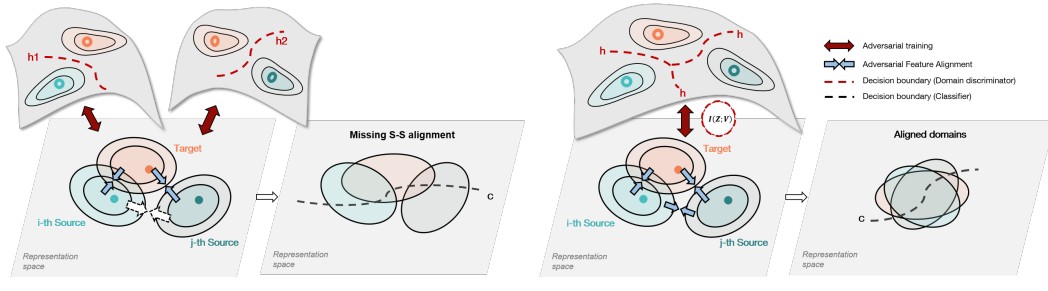

(a) Existing works            (b) Proposed model

Figure 4: Comparison of existing and proposed MDA models. **(a)** Existing multiple-discriminator based methods align each pairwise source and target domain but may fail by neglecting the domain shift between source domains. It also may suffer from unstable optimization and lack of resource-efficiency. **(b)** Our proposed model mitigates suggested problems by unifying domain discriminators.

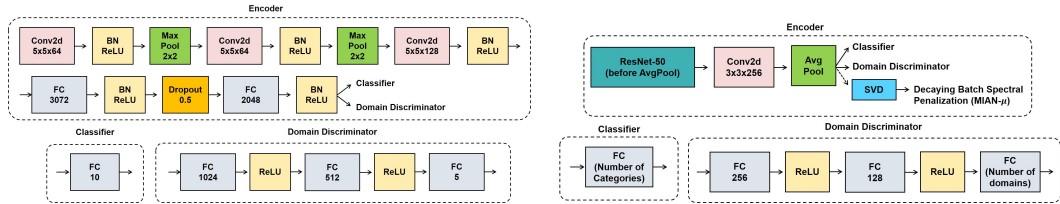

(a) Encoder, domain discriminator, and classifier used in Digits-Five experiments  (b) Encoder, domain discriminator, and classifier used in Office-31 and Office-Home experiments

Figure 5: Network architectures. BN denotes Batch Normalization (Ioffe & Szegedy (2015)) and SVD denotes differentiable SVD in PyTorch for **MIAN-$\gamma$** (Section E)

Batch Spectral Penalization (**BSP**, Chen et al. (2019)), Adversarial Discriminative Domain Adaptation (**ADDA**, Tzeng et al. (2017)), Maximum Classifier Discrepancy (**MCD**, Saito et al. (2018)), Deep Cocktail Network (**DCTN**, Xu et al. (2018)), and Moment Matching for Multi-Source Domain Adaptation (**M$^3$SDA**, Peng et al. (2019)).

Table 4: Experimental setup. The batch size for each domain is reported.

| Dataset | Optimization method | Learning rate | Momentum | Batch size | Iteration |
|---|---|---|---|---|---|
| Digits-Five | Adam | $2e^{-4}$ | (0.9, 0.99) | 128 | 50000 |
| Office-31 | mini-batch SGD | $1e^{-3}$ | 0.9 | 16 | 25000 |
| Office-Home | mini-batch SGD | $1e^{-3}$ | 0.9 | 16 | 25000 |

**Hyperparameters** Details of the experimental setup are summarized in Table 4. Other state-of-the-art adaptation models are trained based on the same setup except for these cases: **DCTN** show poor performance with the learning rate shown in Table 4 for both Office-31 and Office-Home datasets. Following the suggestion of the original authors, $1e^{-5}$ is used as a learning rate with the Adam optimizer (Kingma & Ba (2014)); **MCD** show poor performance for the Office-Home dataset with the learning rate shown in Table 4. $1e^{-4}$ is selected as a learning rate. For both the proposed and other baseline models, the learning rate of the classifier or domain discriminator trained from the scratch is set to be 10 times of those of ImageNet-pretrained weights, in Office-31 and Office-Home datasets. More hyperparameter configurations are summarized in Table 5 (Section E)

## C PSEUDOCODE

Due to the limited space, we provide the algorithm of **MIAN** in this Section. Details about training-dependent scaling of $\beta_t$ are in Section E.

---

**Algorithm 1:** Multi-source Information-regularized Adaptation Networks (**MIAN**)

---

mini-batch size for each domain=$m$, Number of source domains=$N$, Training iteration $T$.
$M=m(N+1)$, Set of domain labels $\mathcal{V} = \{1, \ldots, N+1\}$.
**for** $t \leftarrow 1$ **to** $T$ **do**

    $X = \{\mathbf{x}_i\}_{i=1}^{M}$ is a union of samples $\{X_{S_1}, \ldots, X_{S_N}, X_T\}$
    $Y = \{\mathbf{y}_i\}_{i=1}^{mN}$ is a union of samples $\{Y_{S_1}, \ldots, Y_{S_N}\}$
    Let $\mathbf{z}_i = F(\mathbf{x}_i)$, and $\hat{\mathbf{y}}_i = C(F(\mathbf{x}_i)), \forall \mathbf{x}_i \in X$

    $L(h) = -\frac{1}{M} \sum_{\mathbf{v} \in \mathcal{V}} \sum_{i:\mathbf{v}_i = \mathbf{v}} \left[ \mathbb{1}_{[k=\mathbf{v}_i]}^{T} \log h(\mathbf{z}_i) + \mathbb{1}_{[k \neq \mathbf{v}_i]}^{T} \log(\mathbb{1} - h(\mathbf{z}_i)) \right]$
    Backpropagate gradient of $L(h)$, or the variant (Mao et al. (2017)), to $h$.

    $L(F,C) = -\frac{1}{mN} \sum_{\mathbf{y} \in \mathcal{Y}} \sum_{i:\mathbf{y}_i = \mathbf{y}} \left[ \mathbb{1}_{[k=\mathbf{y}_i]}^{T} \log \hat{\mathbf{y}}_i \right]$
    $\beta_t = \beta_0 \cdot 2 \left( 1 - \frac{1}{1 + exp(-\sigma \cdot t/T)} \right)$                    `// See Appendix E`

    $L(F) = L(F,C) - \beta_t L(h)$
    Backpropagate gradient of $L(F)$ to $F$.
    Backpropagate gradient of $L(F,C)$ to $C$.

---

# D  ADDITIONAL RESULTS

**Visualization of learned latent representations.** We visualized domain-independent representations extracted by the input layer of the classifier with t-SNE (Figure 6). Before the adaptation process, the representations from the target domain were isolated from the representations from each source domain. However, after adaptation, the representations were well-aligned with respect to the class of digits, as opposed to the domain.

**Hyperparameter sensitivity.** We conducted the analysis on hyperparameter sensitivity with degree of regularization $\beta$. The target domain is set as Amazon or Art, where the value $\beta_0$ changes from 0.1 to 0.5. The accuracy is high when $\beta_0$ is approximately between 0.1 and 0.3. We thus choose $\beta_0 = 0.2$ for Office-31, and $\beta_0 = 0.3$ for Office-Home.

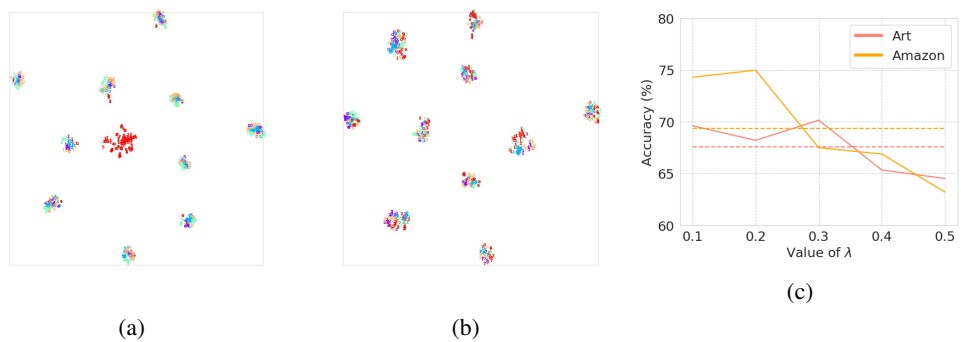

(a)                    (b)                  (c)

Figure 6: **(a)**∼**(b)**: tSNE visualization **(a)** before and **(b)** after adaptation. Representations from target domain (SVHN) are shown in red. Digit class labels are shown with corresponding numbers. **(c)** Analysis on hyperparameter sensitivity.

# E  DECAYING BATCH SPECTRAL PENALIZATION

In this Section, we provides details on the Decaying Batch Spectral Penalization (DBSP) which expands **MIAN** into **MIAN-$\gamma$**.

### E.1 BACKGROUNDS

There is little motivation for models to control the complex mutual dependence to domains if reducing the entropy of representations is sufficient to optimize the value of $I(Z; V) = H(Z) - H(Z \mid V)$. If so, such implicit entropy minimization substantially reduce the upper bound of $I(Z; Y)$, leading to a increase in optimal joint risk $\lambda^*$. In other words, the decrease in the entropy of representations may occur as the side effect of $I(Z; V)$ regularization. Such unexpected side effect of information regularization is highly intertwined with the hidden deterioration of discriminability through adversarial training (Chen et al. (2019); Liu et al. (2019)).

Based on these insights, we employ the SVD-entropy $H_{SVD}(\mathbf{Z})$ (Alter et al. (2000)) of a representation matrix $\mathbf{Z}$ to assess the richness of the latent representations during adaptation, since it is difficult to compute $H(Z)$. Note that while $H_{SVD}(\mathbf{Z})$ is not precisely equivalent to $H(Z)$, $H_{SVD}(\mathbf{Z})$ can be used as a proxy of the level of disorder of the given matrix (Newton & DeSalvo (2010)). In future works, it would be interesting to evaluate the temporal change in entropy with other metrics. We found that $H_{SVD}(\mathbf{Z})$ indeed decreases significantly during adversarial adaptation, suggesting that some eigenfeatures (or eigensamples) become redundant and, thus, the inherent feature-richness diminishes. (Figure 7a) To preclude such deterioration, we employ Batch Spectral Penalization (BSP) (Chen et al. (2019)), which imposes a constraint on the largest singular value to solicit the contribution of other eigenfeatures. The overall objective function in the multi-domain setting is defined as:

$$\min_{F,C} \ L(F,C) + \beta \hat{I}(Z;V) + \gamma \sum_{i=1}^{N+1} \sum_{j=1}^{k} s_{i,j}^2, \tag{28}$$

where $\beta$ and $\gamma$ are Lagrangian multipliers and $s_{i,j}$ is the $j$th singular value from the $i$th domain. We found that SVD entropy of representations is severely deteriorated especially in the early stages of training, suggesting the possibility of over-regularization. The noisy domain discriminative signals in the initial phase (Ganin et al. (2016)) may distort and simplify the representations. To circumvent the impaired discriminability in the early stages of the training, the discriminability should be prioritized first with high $\gamma$ and low $\beta$, followed by a gradual decaying and annealing in $\gamma$ and $\beta$, respectively, so that a sufficient level of domain transferability is guaranteed. Based on our temporal analysis, we introduce the training-dependent scaling of $\beta$ and $\gamma$ by modifying the progressive training schedule (Ganin et al. (2016)):

$$\begin{aligned} \beta_p &= \beta_0 \cdot 2\big(1 - \frac{1}{1 + exp(-\sigma \cdot p)}\big) \\ \gamma_p &= \gamma_0 \cdot \big(\frac{2}{1 + exp(-\sigma \cdot p)} - 1\big), \end{aligned} \tag{29}$$

where $\beta_0$ and $\gamma_0$ are initial values, $\sigma$ is a decaying parameter, and $p$ is the training progress from 0 to 1. We refer to this version of our model as **MIAN-$\gamma$**. Note that **MIAN** only includes annealing-$\beta$, excluding DBSP. For the proposed method, $\beta_0$ is chosen from $\{0.1, 0.2, 0.3, 0.4, 0.5\}$ for Office-31 and Office-Home dataset, while $\beta_0 = 1.0$ is fixed in Digits-Five. $\gamma_0$ is fixed to $\{1e^{-4}\}$ following Chen et al. (2019).

Table 5: Hyper parameters configuration. Annealing-$\beta$ is not adopted in the Digits-Five experiment. Decaying batch spectral penalization is not adopted in the **MIAN**.

| Dataset(Model) | $\beta_0$ | $\gamma_0$ | $\sigma$ | $k$ |
|---|---|---|---|---|
| Digits-Five (**MIAN**) | 1.0 | N/A | N/A | N/A |
| Office-31 (**MIAN**) | 0.1 | N/A | 10.0 | N/A |
| Office-31 (**MIAN-$\gamma$**) | 0.2 | 0.0001 | 10.0 | 1 |
| Office-Home (**MIAN**) | 0.3 | N/A | 10.0 | N/A |
| Office-Home (**MIAN-$\gamma$**) | 0.3 | 0.0001 | 10.0 | 1 |

### E.2 EXPERIMENTS

**SVD-entropy.** We evaluated the degree of compromise of SVD-entropy owing to transfer learning. For this, DSLR was fixed as the source domain, and each Webcam and Amazon target domain

was used to simulate low (DSLR→Webcam; *DW*) and high domain (DSLR→Amazon; *DA*) shift conditions, respectively. SVD-entropy was applied to the representation matrix extracted from ResNet-50 and **MIAN** (denoted as *Adapt* in Figure 7a) with constant $\beta = 0.1$. For accurate assessment, we avoided using spectral penalization. As depicted in the Figure 7a, adversarial adaptation, or information regularization, significantly decreases the SVD-entropy of both the source and target domain representations, especially in the early stages of training, indicating that the representations are simplified in terms of feature-richness. Moreover, when comparing the *Adapt_DA_source* and *Adapt_DW_source* conditions, we found that SVD-entropy decreases significantly as the degree of domain shift increases.

We additionally conducted analyses on temporal changes of SVD entropy by comparing BSP and decaying BSP (Figure 7b). SVD entropy gradually decreases as the degree of compensation decreases in DBSP which leads to improved transferability and accuracy. Thus DBSP can control the trade-off between the richness of the feature representations and adversarial adaptation as the training proceeds.

**Ablation study.** We performed an ablation study to assess the contribution of the decaying spectral penalization and annealing information regularization to DA performance (Table 6, 7). We found that the prioritization of feature-richness in early stages (by controlling $\beta$ and $\gamma$) significantly improves the performance. We also found that the constant penalization schedule (Chen et al. (2019)) is not reliable and sometimes impedes transferability in the low domain shift condition (Webcam, DSLR in Table 6). This implies that the conventional BSP may over-regularize the transferability when the degree of domain shift and SVD-entropy decline are relatively small.

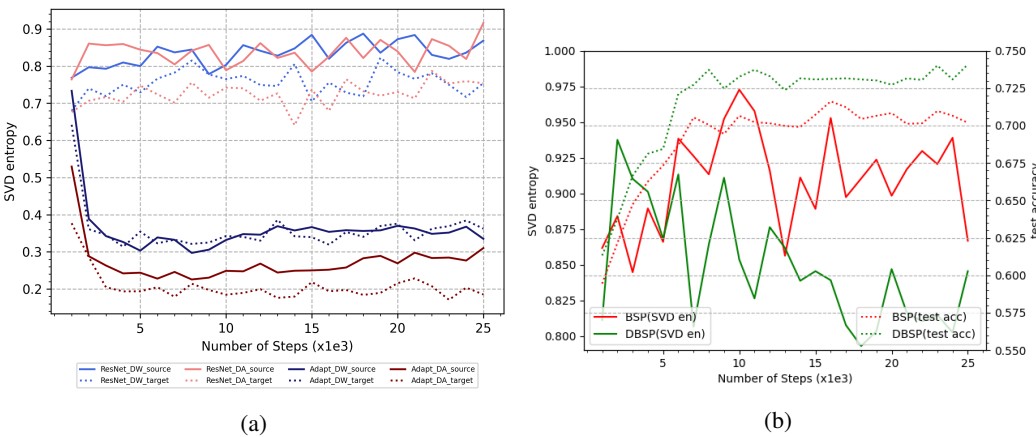

(a)          (b)

Figure 7: **(a)**: SVD-entropy analysis. (Office-31; Source domain: DSLR) **(b)**: Comparisons between BSP and DBSP. (Office-31; DSLR → Amazon)

Table 6: Ablation study of decaying batch spectral penalization and annealing information regularization (Office-31). For accurate assessment of extent to which performance improvement is caused by each strategies, $\gamma$ is fixed as 0 in *Annealing-$\beta$*, and $\beta$ is fixed as 0.1 in *Decaying-$\gamma$*. Results from *Annealing-$\beta$* are reported in main paper.

| Standards | Hyper parameters | Amazon | DSLR | Webcam | Avg |
|---|---|---|---|---|---|
| Baseline | $\beta = 0.1$ as a constant | 69.98 | 99.48 | 98.13 | 89.20 |
| Annealing-$\beta$ | $\beta_0 = 0.1, \sigma = 10$ | **74.65** | **99.48** | **98.49** | **90.87** |
| Decaying-$\gamma$ | $\gamma = 1e^{-4}$ as a constant | 74.73 | 98.65 | 96.24 | 89.87 |
| | $\gamma_0 = 1e^{-4}, \sigma = 10$ | **75.01** | **99.68** | 98.10 | **90.93** |
| Full version | $\beta_0 = 0.1, \gamma_0 = 1e^{-4}, \sigma = 10$ | **76.15** | 99.22 | **98.39** | **91.26** |

Table 7: Accuracy (%) on Office-Home dataset. Results from **MIAN** are reported in main paper.

| Standards | Art | Clipart | Product | Realworld | Avg |
|---|---|---|---|---|---|
| **MIAN** | 69.39±0.50 | 63.05±0.61 | 79.62±0.16 | 80.44±0.24 | 73.12 |
| **MIAN-$\gamma$** | **69.88±0.35** | **64.20±0.68** | **80.87±0.37** | **81.49±0.24** | **74.11** |

