# OpenReview forum: "A Simple Unified Information Regularization Framework for Multi-Source Domain Adaptation"
_ICLR.cc/2021/Conference — Reject_

### Official Review · AnonReviewer1 · 2020-10-26
**This paper aims to propose a new method for multi-source domain adaptation with a strong theoretical flavour but the writing is hard to follow and the experimental comparison needs clarification.**

**Rating:** 4
**Confidence:** 4

**Review:**

This paper proposes a Multi-source Information-regularized Adaptation Networks (MIAN) for multi-source domain adaptation. The presentation has a strong theoretical flavour. MIAN is evaluated on three benchmark datasets against other methods to show its superiority. At times, the paper is difficult to follow and lacks clarity. Please see my detailed comments below.

1. The paper is organised with three pages of theoretical insights (Section 3) after Section 2 on related works, which could be difficult for many readers. It could be better to convey the intuition, high-level idea, or big picture with some visual illustration that can help readers appreciate the proposed idea(s).

2. The paper reproduced quite some existing theories (and also a proof in the  appendix). I am wondering whether these materials can be presented in a more accessible and compact way.

3. Although source code has been provided (which is good), providing pseudo-code can help readers better understand the proposed method and differentiate it from other existing ones.

4. It will be better to perform some computational complexity analysis to give a fuller picture including the efficiency of the proposed method.

5. At the top of Page 7, it says "we reproduced all the other baseline results using the same backbone architecture and optimizer settings as the proposed method". Have you tune the optimizer settings for the proposed method and baselines? How did you determine the hyperparameters reported in the appendix? Grid search?

6. It is not clear why standard deviations are not reported for Digits-Five.

7. On the three datasets studied, the models evaluated seem to be different for different datasets. For example, there is no result on Single-best at all for Office-Home. For "single-best", seven methods (besides source only) were reported for Digits-Five but only DAN and JAN were reported for Office-31.

8. Repeating each experiment for only four times may not give a good estimation of the variance. In addition, the 100.00+-0.00 result of DAN on DSLR is impressive. I am wondering whether this perfect result will hold if you have more repetitions.

9. On Office-31, MIAN actually only outperforms JAN's single-best by 2%, and the second best by 1.4%, which is far from a large margin claimed.

10. Figure 2a and 2b (similarly 2c and 2d) are hardly readable on print.

11. Minor issues.
Figure 1 could be put at the top rather than in the middle of text.
Acronyms are not always defined, e.g. FCN in Figure 1, although knowledgeable readers know what it means.

---

> ### Author Response · Authors · 2020-11-20
> **Thank you for the valuable comments and feedback. [1/2]**
>
> Thank you for insightful feedback. We appreciate you giving practical advice to improve the quality of paper. We are glad to see that you recognize theoretical and empirical contributions of our work to the MDA problem. In the below we provide point-by-point responses to fully clarify the issues (the reviewer’s comment in highlight, followed by our reply).
>
> > The paper is organised with three pages of theoretical insights (Section 3) after Section 2 on related works, which could be difficult for many readers. It could be better to convey the intuition, high-level idea, or big picture with some visual illustration that can help readers appreciate the proposed idea(s).
>
> Thank you very much for the very constructive suggestion! Though we agree that the paper is a bit dense to follow, we would like to respectfully note that the theoretical connection between adversarial domain adaptation and information regularization is not trivial. We thus aimed to establishes the bridge between two different fields step by step; (1) recasting single-source DA as a constrained optimization, (2) establishing a theoretical connection between this problem and information regularization problem, (3) generalizing such findings to the information bottleneck theory, and (4) newly to multi-source domains in the revised version (Section 4).
>
> To clearly provide the big picture and the connection between theoretical components more clearly, we have revised both the abstract and introduction to highlight the following problems: dispersed domain-discriminative knowledge, lack of scalability, and high variance in the objective. To show these problems more clearly, we have added discussion on the relationship between information regularization and $\cal{H}$-divergence optimization given multiple source domains. In Section 4, we have shown that the proposed information regularization can be expanded into minimizing the lower bound of an average of domain discrepancy $\hat d_{\cal{H}}(\cal{V})$ between the given domain and the mixture of remaining domains with a simple modification. Thus our information-theoretical framework is closely related to indirectly minimize the lower bound of every pairwise domain discrepancy with a *single* discriminator. We believe that unifying the discriminator allows us to glean domain-shared information from data and to properly align all domains together. Please refer to Section 4 and appendix Section A for more details. In Section 5, we have also shown that such modification indeed improves the performance and decreases $\hat d_{\cal{H}}(\cal{V})$.
>
>
> Moreover, to fully reflect the reviewer’s suggestion, we have added visual/conceptual illustration of our model to the appendix B.
>
> > Although source code has been provided (which is good), providing pseudo-code can help readers better understand the proposed method and differentiate it from other existing ones.
>
> We agree with you that providing pseudo-code would be helpful for better understanding. We have added it to the revised version of our manuscript (appendix C).
>
> > It will be better to perform some computational complexity analysis to give a fuller picture including the efficiency of the proposed method.
>
> We have added discussion about MIAN’s improved resource efficiency to Section 4. Specifically, we have shown that the proposed information regularization can be extended to minimizing the expectation of domain discrepancy between the given domain and the mixture of remaining domains with a simple modification. Existing approaches require about $\mathcal{O}(N^2)$ number of domain classifiers in order to accurately estimate the same metric. Even if we neglect the discrepancy between source domains, we still need $\mathcal{O}(N)$ number of domain classifiers. In this regard, there is no comparison between the proposed method using a single domain classifier and existing approaches in terms of resource efficiency.

---

> > ### Author Response · Authors · 2020-11-20
> > **Thank you for the valuable comments and feedback. [2/2]**
> >
> > > Have you tune the optimizer settings for the proposed method and baselines? How did you determine the hyperparameters reported in the appendix? Grid search?
> >
> > Since we found that using SGD with momentum outperforms other optimizer settings, e.g., Adam, we fixed the optimizer as SGD with momentum=0.9 for all of the simulations. However, some of the baseline methods (DCTN, MCD) show poor performance with the above optimizer. We basically used the setting suggested by the original authors, but also tried our best to fine-tune the learning rate for the baseline models (although we were not able to run exhaustive grid search as the size of the search space exponentially increases with the number of hyperparameters). Note that unlike these baseline models, MIAN does not require many hyperparameters, which significantly reduces the burden of fine-tuning. For the proposed method, $\beta_0$ is chosen from $\{0.1, 0.2, 0.3, 0.4, 0.5\}$ for Office-31 and Office-Home dataset, while $\beta_0=1.0$ is fixed in Digits-Five. $\gamma_0$ is fixed to $\{1e^{-4}\}$ following the setting of the original BSP. To fully address this issue, we have added the hyperparameter sensitivity study to Appendix D.
> >
> > > It is not clear why standard deviations are not reported for Digits-Five.
> >
> > The baseline results for the Digits-Five dataset were excerpted from ref-1. We were not able to report the standard deviations because, unfortunately, they were not provided in this study. Note that we report the standard deviations of MIAN as belows.
> >
> > |          | MNIST-M          | MNIST            | USPS             | SVHN             | SYNTH            |
> > |----------|------------------|------------------|------------------|------------------|------------------|
> > | Accuracy | 84.36 $\pm$ 0.91 | 97.91 $\pm$ 0.49 | 96.49 $\pm$ 1.00 | 88.18 $\pm$ 0.95 | 93.23 $\pm$ 0.65 |
> >
> > > The models evaluated seem to be different for different datasets. For example, there is no result on Single-best at all for Office-Home. For "single-best", seven methods (besides source only) were reported for Digits-Five but only DAN and JAN were reported for Office-31
> >
> > We apologize for not being able to reproduce all the baseline results mainly due to heavy computational burden. We plan to supplement some of the missing experiments in the future work. For some baseline models, we have done our best to reproduce the results with their official/public implementations, but to no avail; e.g., MDAN.
> >
> > > Repeating each experiment for only four times may not give a good estimation of the variance.
> >
> > We agree that repeating each experiment many times would increase the reliability of the results. We followed the convention of other related works (repeating about 4-5 times; we suspect this is an acceptable compromise accommodating the heavy computational demands). That said, from our preliminary analyses in which simulations were run many times for some cases, we learned that the results are reasonably reliable to the extent to which more simulations would not significantly alter the patterns of the results. We will do our best to supplement them before the camera-ready deadline (if accepted).
> >
> > > On Office-31, MIAN actually only outperforms JAN's single-best by 2%, and the second best by 1.4%, which is far from a large margin claimed.
> >
> > Though the improvement compared to the baselines seems to be limited in some cases, note that the key performance indicator of MDA should be improvement in *difficult task transfer with high domain shift*. The performance improvement by a small margin might be ascribed to the fact that available source domains usually contain sufficient information for task performance. For example, MIAN outperforms other baselines by +12.0% in MNIST-M, +5.2% in Amazon and +2.4% in Art domains, all of which falls in the category of difficult target domains. Moreover, since MIAN significantly decreases the variance of stochastic gradients by reducing multiple discriminators to unified one, we expect that MIAN will outperform other baselines by a larger margin if the number of domains increases. To avoid any misunderstanding, we tone it down by saying “by a large margin especially for difficult target domains.”
> >
> > > Figure 2a and 2b (similarly 2c and 2d) are hardly readable on print.
> >
> > Thank you. We have increased the size of the fonts/legends, as well as improving the resolution of figures in the revised paper.
> >
> > > Figure 1 could be put at the top rather than in the middle of text. Acronyms are not always defined, e.g. FCN in Figure 1, although knowledgeable readers know what it means.
> >
> > Thank you for the constructive comments, which would greatly improve the clarity of Fig1. We have revised this figure and caption as advised.

---

### Official Review · AnonReviewer4 · 2020-10-27
**A simple, powerful multi-source information regularized domain adaptation framework**

**Rating:** 7
**Confidence:** 3

**Review:**

This paper studies the multi-source domain adaptation problem. The authors examine the existing MDA solutions, i.e. using a domain discriminator for each source-target pair, and argue that the existing ones are likely to distribute the domain-discriminative information across multiple discriminators. By theoretically analyzing from the information regularization point, the authors present a simple yet powerful architecture called multi-source information-regularized adaptation network, MIAN.

I vote to accept the paper.
-This paper has a clear motivation, is well written, and establishes the final objective step by step with the theoretical supports.

-The proposed objective is simple but powerful. I enjoy reading the analysis of advantages over the existing solutions, which is well reflected in the experiments. The reported performance is competitive, even compared to the missing reference ECCV’20 (see below).

-The quantitative analyses validate the effectiveness of the model design. Particularly, the analysis on variance of stochastic gradients validates the technical benefits on optimization stability.

A few comments/suggestions.
-A few recent MDA works are missing, including but not limited to [ref-1, ref-2, ref-3]. Although not all of them using image datasets as testbed, but I would encourage the authors to include and discuss them under the same structure.
[ref-1] Hang Wang et al., Learning to Combine: Knowledge Aggregation for Multi-Source Domain Adaptation, ECCV 2020
[ref-2] Chuang Lin et al., Multi-source Domain Adaptation for Visual Sentiment Classification, https://arxiv.org/abs/2001.03886
[ref-3] Haotian Wang et al., Tmda: Task-specific multi-source domain adaptation via clustering embedded adversarial training. ICDM 2019.

-In section 3.3, the authors argue that their frameworks filtering out domain-specific information while preserving the amount of domain-shared information. The statement and the earlier discussion are intuitively correct to me. However, it would be great to see a quantitative or qualitive study on the effectiveness of the preserved domain-shared information and domain-specific separately.

-Minors. In page 5, “It bias the representation towards …” -> “biases”

=====
Updates: Thanks for the authors' response. I carefully read other reviewers' comments and responses. My concern on the missing study of empirical or theoretical support of claim "framework can filter out domain-specific information while preserving the amount of domain-shared information" was also raised by other reviewer. Overall, I still believe this paper provides new insights for this field.

---

> ### Author Response · Authors · 2020-11-20
> **Thank you for the valuable comments and feedback.**
>
> Thank you for insightful comments and feedback. We are happy to hear that the reviewer found our work is well motivated and provides sufficient theoretical justifications and evidence. In the below, we provide point-by-point responses to the reviewer’s comments (the reviewer’s comment in highlight, followed by our reply).
> > A few recent MDA works are missing, including but not limited to [ref-1, ref-2, ref-3]. Although not all of them using image datasets as testbed, but I would encourage the authors to include and discuss them under the same structure.
>
> The suggested works are closely related to our work in that they highlight the importance of domain knowledge aggregation. Hang Wang et al., 2020 proposes a Learning to Combine for Multi-Source Domain Adaptation (LtC-MSDA) framework, which encourages alignment and interaction of representations from different domains. Haotian Wang et al., 2019 stressed out that the dispersed multi-source domain representations may disturb the adversarial learning process and propose the clustering embedded adversarial training method. While the explicit objectives may not coincide and each with a specific application, our work proposes a unified information theoretical framework to formally demonstrate the importance of such aggregation. We have consolidated the claim on the importance of aggregating domain-discriminative information below. Also, we will compare and discuss the references in the revised version of our manuscript and future work if possible.
>
> > (...) The statement and the earlier discussion are intuitively correct to me. However, it would be great to see a quantitative or qualitive study on the effective of the preserved domain-shared information and domain-specific separately.
>
> We agree that some part of our claim could be improved by supplementing experimental evidence. Although quantifying the domain-shared/domain-private information is not trivial, we still believe that the multiple discriminators inevitably distribute valuable domain-shared information. To show these problems more clearly, we have added discussion on the relationship between information regularization and $\cal{H}$-divergence optimization given multiple source domains. In Section 4, we have shown that the proposed information regularization can be expanded into minimizing the lower bound of an average of domain discrepancy $\hat d_{\cal{H}}(\cal{V})$ between the given domain and the mixture of remaining domains with a simple modification. Thus our information-theoretical framework is closely related to indirectly minimize the lower bound of every pairwise domain discrepancy with a *single* discriminator. We believe that unifying the discriminator allows us to glean domain-shared information from data and to properly align all domains together. Please refer to Section 4 and appendix Section A for more details. In Section 5, we have also shown that such modification indeed improves the performance and decreases $\hat d_{\cal{H}}(\cal{V})$.
>
> To summarize, we provided both theoretical arguments and empirical evidence to fully justify three potential pitfalls of using multiple discriminators: dispersed domain-discriminative knowledge, lack of scalability, and high variance in the objective. We propose that such concerns can be eased by reducing multiple discriminators to unified one both theoretically and empirically.

---

### Official Review · AnonReviewer3 · 2020-10-29
**strong motivation, but limited contribution and unclear explanation**

**Rating:** 5
**Confidence:** 4

**Review:**

This paper proposes a simple unified information regularization framework for multi-source domain adaptation. Experimental results verify the effectiveness of the proposed method.

The paper is well organized. The motivation is clear and the idea of reducing multiple discriminators to single discriminator is very reasonable. The experiments are somewhat convincing.

The main weaknesses of this paper lie in the following aspects:

- The authors claim that domain-discriminative knowledge from other domains is not fully leveraged by multiple individual discriminators, while their proposed MIAN can filter out domain-specific information while preserve domain-shared information. However, such a claim is not supported by solid theoretical or empirical evidence but only the authors intuition.  Since the motivation of the whole paper is based on this claim, it greatly undermines the basis of this paper. Hence, it is critical for the authors to provide sufficient evidence to demonstrate this claim.

- The overall contribution is not that large, as  the proposed information regularization method is simply based on recently proposed related work.  And some aspects of the proposed method are not clearly explained. For example, why Eqn. (3) and Eqn. (6) are equivalent? It is not very obvious to me.

- The experimental results are not good enough. The proposed MIAN exhibits good performance on small datasets like Digits-Five and Office-31. But its performance (an average of 73.12) on larger dataset like Office-Home is worse than other competitors like MFSAN (an average of 74.1). Furthermore, why not conduct experiments on DomainNet, the largest benchmark created specifically for DA?

- The experimental analysis seems to be inadequate. The paper lacks hyperparameter sensitivity analysis, which is standard practice in UDA area. Furthermore, although there are some analysis on variance of stochastic gradients, there is no evidence to support that this will lead to better performance, or  increased computational-efficiency or improved stability.

- The presentation can be greatly improved. Some typos exist and some mathematical equations are not formal enough. E.g., $\Xcal$, $\Zcal$ and $M$ are used without or before definition.

---

> ### Author Response · Authors · 2020-11-20
> **Thank you for the valuable comments and feedback. [1/2]**
>
> Thank you for valuable feedback. We appreciate you giving a fruitful comment on the validity of the claim / additional experiments. We are happy to hear that you found our work well-organized and motivated. In the below, we provide point-by-point responses to fully clarify the issues (the reviewer’s comment in highlight, followed by our reply).
>
> > The authors claim that domain-discriminative knowledge from other domains is not fully leveraged by multiple individual discriminators, (...) However, such a claim is not supported by solid theoretical or empirical evidence but only the authors intuition.
>
> Sorry that we did not make clear these points. Although quantifying the domain-shared/domain-private information is not trivial, we argue that the multiple discriminators inevitably distribute valuable domain-shared information, as supported by our theoretical framework. That being said, to illustrate these problems more clearly, we have added discussion on the relationship between information regularization and $\cal{H}$-divergence optimization given multiple source domains. In Section 4, we have shown that the proposed information regularization can be expanded into minimizing the lower bound of an average of domain discrepancy $\hat d_{\cal{H}}(\cal{V})$ between the given domain and the mixture of remaining domains with a simple modification. Thus our information-theoretical framework is closely related to indirectly minimize the lower bound of every pairwise domain discrepancy with a *single* discriminator. We believe that unifying the discriminator allows us to glean domain-shared information from data and to properly align all domains together. Please refer to Section 4 and appendix Section A for more details. In Section 5, we have also shown that such modification indeed improves the performance and decreases $\hat d_{\cal{H}}(\cal{V})$.
>
> To summarize, we provided both theoretical arguments and empirical evidence to fully justify three potential pitfalls of using multiple discriminators: dispersed domain-discriminative knowledge, lack of scalability, and high variance in the objective. We propose that such concerns can be eased by reducing multiple discriminators to a unified one both theoretically and empirically.
>
> > The overall contribution is not that large, (...) And some aspects of the proposed method are not clearly explained. For example, why Eqn. (3) and Eqn. (6) are equivalent?
>
> While the proposed regularization method is based on the previous derivations, we respectfully note that the proposed framework has considerable merit in both theoretical and empirical aspects. To the best of our knowledge, this is the first work to afford a bridge between adversarial domain adaptation and information bottleneck theory, which can also be generalized to multiple domains. Moreover, our proposed framework opens up new possibilities for various machine learning problems, e.g., information regularization with stochastic domain labels, recognizing the importance of alignment of source domains for adversarial adaptation, or improving the proposed regularizer by introducing domain weights.
>
> We agree with you that eq. (3) and eq. (6) is not strictly equivalent due to the logarithm. To avoid any confusion, we toned down the corresponding phrase to “close relationship”. While these equations are not equivalent, we believe that eq. (3) can be viewed as a differentiable version of eq. (6), which can be applied as a practical objective. In Section 4, we have supplemented remarks on the advantage of our formulation: “Since the output vector $h(\mathbf{z})$ in (15) often comes from the $\arg \max$ operation, (15) is not differentiable w.r.t. $\mathbf{z}$. However, our framework has a differentiable objective as in (14).”
>
> > But its performance (an average of 73.12) on larger dataset like Office-Home is worse than other competitors like MFSAN (an average of 74.1). Furthermore, why not conduct experiments on DomainNet, the largest benchmark created specifically for DA?
>
> Thanks for the useful suggestion. We would like to point out that MIAN has been fairly compared to other baseline models with fixed experimental settings. For example, while the suggested paper has shown great performance, some of the hyperparameters are not consistent with the ones used in our experiment. For example, MFSAN uses a larger batch size or deeper bottleneck architecture. DBSN trains the model with much longer iteration, as suggested by reviewer 2. After fixing the experimental settings, we have conducted additional experiments, and confirmed that MIAN outperformed both MFSAN and DBSN! Also, we appreciate the reviewer recommending the new dataset. We will conduct experiments on DomainNet in the future work.

---

> > ### Author Response · Authors · 2020-11-20
> > **Thank you for the valuable comments and feedback. [2/2]**
> >
> > |       | Art                | Clipart            | Product            | Realworld         | Avg   |
> > |-------|--------------------|--------------------|--------------------|-------------------|-------|
> > | MFSAN | 69.21 $\pm$ 0.89  | 60.77 $\pm$ 0.20  | 77.68 $\pm$ 0.01  | **80.65 $\pm$ 0.09** | 72.08 |
> > | MIAN  | **69.39 $\pm$ 0.50** | **63.05 $\pm$ 0.61** | **79.62 $\pm$ 0.16** | 80.44 $\pm$ 0.24 | **73.12** |
> >
> > |      | Amazon             | Webcam             | DSLR               | Avg   |
> > |------|--------------------|--------------------|--------------------|-------|
> > | DSBN | 66.82 $\pm$ 0.35  | 94.00 $\pm$ 0.38  | 97.45 $\pm$ 0.22  | 86.09 |
> > | MIAN | **74.65 $\pm$ 0.48** | **99.48 $\pm$ 0.35** | **98.49 $\pm$ 0.59** | **90.87** |
> >
> > > (...) The paper lacks hyperparameter sensitivity analysis, which is standard practice in UDA area. Furthermore, although there are some analysis on variance of stochastic gradients, there is no evidence to support that this will lead to better performance, (...)
> >
> > Thank you for the suggestion. To fully address this issue, we have conducted the hyperparameter sensitivity analysis in Appendix D. Regarding the impact of the variance of stochastic gradients on performance, there have been a lot of interesting works studying the trade-off between fast computation per iteration and slow convergence in SGD, e.g., SVRG, SAG, SAGA (Johnson et al., NIPS 2013, Schmidt et al., 2015, Defazio et al., NIPS 2014): If the variance of the gradient is too large, the convergence speed of SGD would be remarkably slow. For example, SVRG reduces the variance of gradients by keeping a snapshot of the trained model. We also have empirically shown that the convergence speed of *multi_D* is relatively slow compared to MIAN in Figure 2a,b which is consistent with the results from Figure 2c,d.
> >
> > > The presentation can be greatly improved. Some typos exist and some mathematical equations are not formal enough.
> >
> > Thank you. We have fixed some typos and notations as suggested.

---

### Official Review · AnonReviewer2 · 2020-10-30
**A review on a simple unified information regularization framework for multi-source domain adaptation**

**Rating:** 4
**Confidence:** 5

**Review:**

**Summarize what the paper claims to contribute.**
The paper introduces a multi-source information-regularized adaptation network (MIAN). MIAN consists of three parts; information regularization, source classification and batch spectral penalization. The information regularization in MIAN provides an information-theoretic interpretation of multi-source domain adaptation. This interpretation seems to have lower information variance compared to the other multi-source domain adaptation technique relying on multiple binary domain classifiers. The performance of MIAN is evaluated on three benchmarks and it achieves competitive performance comparable to state-of-the-art methods.

**List strong and weak points of the paper.**

*strong points*

The proposed information-theoretic regularization is well-motivated and profound compared to the previous works on multi-source domain adaptation.
Give a first attempt for theoretical explanation on multi-source domain adaptation by extending previous approaches dealing with only source and target domain cases.

*weak points*

A derivation, an ablation study and an analysis on the proposed method are missing.
The paper seems to be not finished yet.

**Clearly state your recommendation (accept or reject) with one or two key reasons for this choice**

I give “Ok but not good enough - rejection (4)” to this paper. The motivation of the paper is plausible to multi-source domain adaptation and the proposed regularization technique seems to be profound. But the supporting evidence of the proposed method, such as appropriate derivation and proof, ablation study and analysis, are missing in the paper. I conjecture that the paper is not finished yet.

**Ask questions you would like answered by the authors to help you clarify your understanding of the paper and provide the additional evidence you need to be confident in your assessment.**

*variance of information*

“In contrast, the variance of our constraint (5) is inversely proportional to $(N+1)^2$” I can’t find the derivation of this term, which is a crucial statement for the proposed method. In the supplementary materials of this paper and (Roh et al., 2020), i could not find the related covariance terms. Could you explain the variance of $\hat{I}(Z; V)$ in eq (5) and $\sum_{k=1}^{N}{\hat{I} (Z_k ; U_k) }$ in terms of the number of domains $N$?

*ablation studies on MIAN*

Which one is the counterpart of MIAN? I want to know the performance of the baseline architecture without MIAN regularization, which is an empirical evidence to check whether MIAN is working or not.

**Provide additional feedback with the aim to improve the paper.**

*paper editing issues*

The methods in the table are not cited in both the table and the main text. This makes it hard to read the experiment table.

*Missing related works.*

Refer [R1, R2] for multi-source domain adaptation. The paper [R2] achieves similar performance with MIAN on multi-source domain adaptation on office-31 benchmark, although it doesn't report standard deviation.
The major works on multi-source domain settings are dealing with domain generalization and larger scale benchmarks, such as DomainNet.

*analysis on the method*

I think that the motivation of this paper can be intuitively explained by using a figure description of the Venn diagram of information-theoretic measures. Refer to the Venn diagram figures on the page [R3].
Is it possible to quantify the information shared among/between domains? I think that kind of measurement is very effective for visualizing multi-source domain adaptation & domain generalization.

[R1] Boosting Domain Adaptation by Discovering Latent Domain, CVPR 18

[R2] Domain Specific Batch Normalization for Unsupervised Domain Adaptation, CVPR 2019

[R3] https://en.wikipedia.org/wiki/Information_theory_and_measure_theory

---

> ### Author Response · Authors · 2020-11-20
> **Thank you for the valuable comments and feedback. [1/2]**
>
> Thank you for the valuable comments and feedback. We are happy to hear that you found our work well-motivated and profound. We greatly appreciate you providing us constructive comments about insufficient supporting evidence, ablation study, and analysis. We fully clear up the issues as stated below (the reviewer’s comment in highlight, followed by our reply).
>
> > But the supporting evidence of the proposed method, such as appropriate derivation and proof, ablation study and analysis, are missing in the paper. I conjecture that the paper is not finished yet. (...) Could you explain the variance of $\hat{I}(Z; V)$  in eq (5) and $\sum{I(Z_k; U_k)}$ in terms of the number of domains $N$?
>
> Our apologies for not making clear the link between our theoretical contributions, and the proposed model derived from our theory, and the ablation study for empirical validation, which caused a misunderstanding. We formulated conventional domain-adaptation as an information-regularization problem, which aims to obtain a domain-independent representation. In doing so, we proposed two bridging theorems (Theorems 2, 3), and the full proofs are provided in the appendix.
>
> Moreover, one of our contributions we would like to note is revisiting the existing approaches based on the proposed framework to unveil the potential risk of high variance in the objective function. As the derivation of $\hat{I}(Z; V)$ may not sufficiently trivial, we add details and derivation in the revised paper eq. (10):
> $Var\Big[{\hat{I}}(Z; V) \Big] = \frac{1}{m^2(N+1)^2} \Big(Var[I_m]\Big),$
> where $I_m$ represents the maximization term in eq. (5) except for the coefficient. Note that the variance of the proposed regularizer is inversely proportional to $(N+1)^2$ in contrast to eq. (9).
> Suppose that a large number of source domains are given, e.g., recorded data from autonomous driving cars, or medical records from various patients. Then since the existing approaches employ multiple domain discriminators for each available source domain, the covariance between the estimated information inevitably arises. We suspect that the covariance will greatly undermine the stability of training since it does not decrease as the number of domains increases. We have shown the experimental evidence in Section 5.
>
> The proposed model, MIAN, is directly built on our theoretical framework (Section 3). In addition to rigorous performance comparison analyses on various benchmark datasets, to assess the degree of contribution of key parts of the proposed model on MDA performance, we did conduct an ablation study, showing that MIAN consistently outperformed another version, in which a unified discriminator is replaced with multiple ones, in terms of accuracy, convergence speed, and variance of gradients! Moreover, we included additional ablation studies on the objective of the domain discriminator. Please refer to Section 4, 5 and our reply to your last comment.
>
> > Which one is the counterpart of MIAN? I want to know the performance of the baseline architecture without MIAN regularization, which is an empirical evidence to check whether MIAN is working or not.
>
> Thanks for the constructive comments. One key factor of MIAN is solving MDA with a unified discriminator, the comparison should be focused on testing the version with multiple discriminators as the baseline model. Therefore, we examined the validity of unifying discriminators by pitting our proposed model, MIAN, against its modified version, multi_D which implements multiple discriminators. (Figure 2) For a completely fair comparison, any other experimental settings, such as base architecture, optimizer, learning rate, and the other hyper-parameters are fixed except the number of domain discriminators. Our ablation study showed that MIAN consistently outperformed multi_D in terms of accuracy, convergence speed, and variance of gradients. Among the existing models, DCTN is one of the closest counterparts which relies on k-way domain discriminators and classifiers. Again, MIAN outperformed DCTN in all three benchmark datasets.
>
> > The methods in the table are not cited in both the table and the main text. This makes it hard to read the experiment table.
>
> We apologize for missing citations of the baseline methods we used in the main text. Due to the limited space, we cited the methods in the appendix, and we have added guidance in the revised paper.

---

> > ### Author Response · Authors · 2020-11-20
> > **Thank you for the valuable comments and feedback. [2/2]**
> >
> > > Refer [R1, R2] for multi-source domain adaptation. The paper [R2] achieves similar performance with MIAN on multi-source domain adaptation on office-31 benchmark, although it doesn't report standard deviation. The major works on multi-source domain settings are dealing with domain generalization and larger scale benchmarks, such as DomainNet.
> >
> > Thank you for the constructive comment, and we agree that dealing with domain generalization or larger scale benchmarks should be one of the future challenges. As mentioned in the paper, although the domain labels are assigned without randomness for simplicity, the domain label also can be generally treated as a stochastic latent random variable in our framework, in line with various related works (Hoffman et al., ECCV 2012, Gong et al., NIPS 2013, Mancini et al., CVPR 2018, Gong et al., CVPR 2019). It is because our information regularizer can address such stochastic domain labels. For example, one may add noise to the domain label as done in Mix-up (Zhang et al., ICLR 2018) to improve the domain-generalizability. In order to generalize the model to the case of infinitely many data (and domains), we believe that expanding our work beyond the discrete domain label will be an interesting future work.
> >
> > We also respectfully note that the performance of domain adaptation models should be tested with a fair experimental environment. For example, while the work the reviewer mentioned has shown great performance, some of the hyperparameters are not consistent to the ones used in our experiment, e.g., larger batch size or longer training iterations. After fixing the experimental settings, we have conducted additional experiments, and found that MIAN outperformed DSBN by a *large* margin (4.78% in average), even though the training of DSBN took two times longer than that of MIAN due to the DSBN’s self-training phase.
> >
> > |      | Amazon             | Webcam             | DSLR               | Avg   |
> > |------|--------------------|--------------------|--------------------|-------|
> > | DSBN | 66.82 $\pm$ 0.35  | 94.00 $\pm$ 0.38  | 97.45 $\pm$ 0.22  | 86.09 |
> > | MIAN | 74.65 $\pm$ 0.48 | 99.48 $\pm$ 0.35 | 98.49 $\pm$ 0.59 | 90.87 |
> >
> > > Is it possible to quantify the information shared among/between domains? I think that kind of measurement is very effective for visualizing multi-source domain adaptation & domain generalization.
> >
> > Thanks for the insightful suggestion. In the revised version of our manuscript, we presented the potential pitfall of dispersing domain-discriminative information with quantitative evidence. Although quantifying the domain-shared/domain-private information is not trivial, we believe that the multiple discriminators inevitably distribute valuable domain-shared information. To illustrate these problems more clearly, we have added discussion on the relationship between information regularization and $\cal{H}$-divergence optimization given multiple source domains. In Section 4, we have shown that the proposed information regularization can be expanded into minimizing the lower bound of average of domain discrepancy $\hat d_{\cal{H}}(\cal{V})$ between the given domain and the mixture of remaining domains with simple modification. Thus our information-theoretical framework is closely related to indirectly minimize the lower bound of every pairwise domain discrepancy, with a *single* discriminator. Please refer to Section 4 and appendix Section A for more details. We believe that unifying the discriminator allows us to glean domain-shared information from data and to properly align all domains together. In Section 5, we have also shown that such modification indeed improves the performance and decreases $\hat d_{\cal{H}}(\cal{V})$.

---

### Author Response · Authors · 2020-11-20
**General response**

We first thank all the reviewers for giving valuable and constructive feedback. We also appreciate positive comments made by reviewers who recognize the theoretical and empirical contributions of our work to the MDA problem. We updated the paper, mainly including:

(1) Revise both the abstract and introduction to highlight the following problems: dispersed domain-discriminative knowledge, lack of scalability, and high variance in the objective.

(2) Add details and derivation of the variance of objective in eq. (10)

(3) Discussion on the relationship between information regularization and $\cal{H}$-divergence
optimization given multiple source domains (Section 4)

(4) Ablation study on the objective function of domain discriminator (Figure 3b,3c)

(5) Lemma 2 (appendix A, Section 4) for the added discussion in (3)

(6) Analysis on Proxy $\mathcal{A}$-distance and empirical mutual information (Figure 3a, 3d)

(7) Visual/conceptual illustration of our model in appendix B

(8) Pseudocode in appendix C

(9) Analysis of hyperparameter sensitivity (Figure 6 in appendix D)

(10) Fix some typos, notations, and figures.

We hope the responses address the concerns of all the reviewers. Also please refer to the updated response for the reviews. More discussion and suggestions are very welcomed.

---

### Decision · Program_Chairs · 2021-01-07
**Final Decision**

**Decision:**

Reject

**Comment:**

Although all reviewers agree that the work is interesting and has potential, several issues in the presentation and the experimental section (especially regarding the ablation) need to be worked on before granting acceptance to the paper.